



**Seasonal and diurnal variations of methane and carbon dioxide in the Kathmandu Valley**
**in the foothills of the central Himalaya**
Khadak Singh Mahata[1,2], Arnico Kumar Panday [3,4], Maheswar Rupakheti[1,5*], Ashish Singh[1],
Manish Naja[6], Mark G. Lawrence[1,2]
[1] Institute for Advanced Sustainability Studies (IASS), Potsdam, Germany
[2] University of Potsdam, Potsdam, Germany
[3] International Centre for Integrated Mountain Development (ICIMOD), Lalitpur, Nepal
[4] University of Virginia, Virginia, USA
[5] Himalayan Sustainability Institute (HIMSI), Kathmandu, Nepal
[6] Aryabhatta Research Institute of Observational Sciences (ARIES), Nainital, India
*Correspondence to: M. Rupakheti (maheswar.rupakheti@iass-potsdam.de)
**Abstract**
The SusKat-ABC (Sustainable Amosphere for the Kathmandu Valley- Atmospheric Brown
Clouds) international air pollution measurement campaign was carried out during December
2012-June 2013 in the Kathmandu Valley and surrounding regions in Nepal. The Kathmandu
Valley is a bowl-shaped basin with a severe air pollution problem. This paper reports
measurements of two major greenhouse gases (GHGs), methane ($CH_4$) and carbon dioxide
($CO_2$), that begun during the campaign and extended for a year at the SusKat-ABC's supersite in
Bode, a semi-urban location in the Kathmandu Valley. Measurements were also made at a
nearby rural site (Chanban), ~25 km (aerial distance) to the southwest of Bode, on the other side
of a tall ridge. The ambient mixing ratios of methane ($CH_4$), carbon dioxide ($CO_2$), water vapor,
and carbon monoxide (CO) were measured with a cavity ring down spectrometer (Picarro
G2401, USA), along with meteorological parameters for a year (March 2013 - March 2014).



Simultaneous measurements were also made at Chanban from 15 July to 3 October 2015. These
measurements are the first of their kind in the central Himalayan foothills. At Bode, the annual
average mixing ratios of $CO_2$ and $CH_4$ were 419.4($\pm$23.9) ppm and 2.193($\pm$0.224) ppm,
respectively. These values are higher than the levels observed at background sites such as Mauna
Loa, USA ($CO_2$: 396.8 ppm, $CH_4$: 1.831 ppm) and Waliguan, China ($CO_2$: 397.7 ppm, $CH_4$:
1.879 ppm) during the same period, and at other urban/semi-urban sites in the region such as
Ahmedabad and Shadnagar (India) and Nanjing (China). They varied slightly across the seasons
at Bode, with seasonal average $CH_4$ mixing ratios being 2.157($\pm$0.230) ppm in the pre-monsoon
season, 2.199($\pm$0.241) ppm in the monsoon, 2.210($\pm$0.200) ppm in the post-monsoon, and
2.214($\pm$ 0.209) ppm in the winter season. The average $CO_2$ mixing ratios were 426.2($\pm$25.5) ppm
in pre-monsoon, 413.5($\pm$24.2) ppm in monsoon, 417.3($\pm$23.1) ppm in post-monsoon, and
421.9($\pm$20.3) ppm in winter season. The maximum seasonal mean mixing ratio of $CH_4$ in winter
was only 0.057 ppm or 2.6% higher than the seasonal minimum during the pre-monsoon period,
while $CO_2$ was 12.8 ppm or 3.1% higher during the pre-monsoon period (seasonal maximum)
than during the monsoon (seasonal minimum).  On the other hand, the CO mixing ratio at Bode
was 191% higher during the winter than during the monsoon season. The enhancement in $CO_2$
mixing ratios during the pre-monsoon season is associated with additional $CO_2$ emissions from
forest fire and agro-residue burning in northern South Asia in addition to local emissions in the
Kathmandu Valley. Published $CO/CO_2$ ratios of different emission sources in Nepal and India
were compared with the observed $CO/CO_2$ ratios in this study. This comparison indicated that
the major sources in the Kathmandu Valley were residential cooking and vehicle exhaust in all
seasons except winter. In winter, the brick kiln emissions were a major source. Simultaneous
measurement in Bode and Chanban  (15 July-3 Oct 2015) revealed that the mixing ratio of $CO_2$,
$CH_4$ and CO mixing ratios were 3.8%, 12%, and 64% higher in Bode than Chanban. Kathmandu
Valley, thus, has significant emissions from local sources, which can also be attributed to its
bowl shaped geography that is conducive to pollution build-up. All three gas species in Bode
showed strong diurnal patterns, whereas $CH_4$ and CO at Chanban did not show any noticeable
diurnal variations.



These measurements provide the first insights into diurnal and seasonal variation of key
greenhouse gases and air pollutants and their local and regional sources, which are important
information for the atmospheric research in the region.

## 1    Introduction


The average atmospheric mixing ratios of two major greenhouse gases (GHGs), $CO_2$ and $CH_4$,
have increased by about 40% (from 278 to 390.5 ppm) and  about 150%  (from 722 to 1803 ppb)
respectively since pre-industrial times (~1750 AD). This is mostly attributed to anthropogenic
emissions (IPCC, 2013). The current global annual rate of increase of the atmospheric $CO_2$
mixing ratio is 1-3 ppm, with average annual mixing ratios now exceeding a value of 400 ppm at
the background reference location in Mauna Loa (WMO, 2016). Between 1750 and 2011,
555(±85) PgC of anthropogenic $CO_2$ was added to the atmosphere, of which two thirds were
contributed by fossil fuel combustion and cement production, with the remaining coming from
deforestation and land use/land cover changes (IPCC, 2013). $CH_4$ is the second largest gaseous
contributor to anthropogenic radiative forcing after $CO_2$ (Forster et al., 2007). The major
anthropogenic sources of atmospheric $CH_4$ are rice paddies, ruminants and fossil fuel use,
contributing approximately 60% to the global $CH_4$ budget (Chen and Prinn, 2006; Schneising et
al., 2009). The remaining fraction is contributed by biogenic sources such as wetlands and
fermentation of organic matter by microbes in anaerobic conditions (Conrad, 1996).
Increasing atmospheric mixing ratios of $CO_2$ and $CH_4$ and other GHGs and short-lived climate-
forcing pollutants (SLCPs) such as black carbon (BC) and tropospheric ozone ($O_3$) have caused
the global mean surface temperature to increase by 0.85°C from 1880 to 2012. The surface
temperature is expected to increase further by up to 2 degrees at the end of the 21[st] century in
most representative concentration pathways (RCP) emission scenarios (IPCC, 2013). The
increase in surface temperature is linked to melting of glaciers and ice sheets, sea level rise,
extreme weather events, loss of biodiversity, reduced crop productivity, and economic losses
(Fowler and Hennessy, 1995; Guoxin and Shibasaki, 2003).
Seventy percent of global anthropogenic $CO_2$ is emitted in urban areas (Fragkias et al., 2013).
Developing countries may have lower per capita GHG emissions than developed countries, but





the large cities in developing countries, with their high population and industrial densities, are
major consumers of fossil fuels and thus, emitters of GHGs. South Asia, a highly populated
region with rapid  growth in urbanization, motorization, and industrialization in recent decades,
has an ever increasing fossil fuel demand and its combustion emitted 444 Tg C/year in 2000
(Patra, et al., 2013), or about 5% of the global total $CO_2$ emissions. Furthermore, a major
segment of the population in South Asia has an agrarian economy and uses biofuel for cooking
activities, which is an important major source of air pollutants and greenhouse gases in the
region.
The emission and uptake of $CO_2$ and $CH_4$ follow a distinct cycle in South Asia.  Ecosystem and
inversion models show that the highest $CO_2$ release to the atmosphere occurs around April and
May while the highest uptake occurs between July-October (Prasad et al., 2014). Patra et al.
(2011) also showed that uptake peaks in August, using an inversion constrained by regional
measurements from commercial aircraft. The observed trend is linked with the growing seasons.
Agriculture is a major contributor of methane emission. For instance, in India it contributes to
75% of $CH_4$ emissions (MoEF, 2007).  Ambient $CH_4$ concentrations are highest during June to
September (peaking in September) in South Asia which are also the growing months for rice
paddies (Goroshi et al., 2011). The minimum ambient $CH_4$ concentrations are observed in
February-March (Prasad et al., 2014).
Climate change has impacted South Asia in several ways, as evident in temperature increase,
change in precipitation patterns, higher incidence of extreme weather events (floods, droughts,
heat waves, cold waves), melting of snowfields and glaciers in the mountain regions, and
impacts on ecosystems and livelihoods (ICIMOD, 2009; MoE, 2011). Countries such as Nepal
are vulnerable to impacts of climate change due to inadequate preparedness for adaptation to
impacts of climate change (MoE, 2011). Decarbonization of its economy can be an important
policy measure in mitigating climate change. Kathmandu Valley is one of the largest
metropolitan cities in the foothills of the Hindu Kush-Himalaya which has significant reliance on
fossil fuels and biofuels. In 2005, fossil fuel burning accounted for 53 % of total energy
consumption in the Kathmandu Valley, while biomass and hydroelectricity were 38% and 9%,
respectively (Shrestha and Rajbhandari, 2010). Fossil fuel consumed in the Kathmandu Valley



accounts for 32% of the country's fossil fuel imports, and the major fossil fuel consumers are
residential (53.17%), transport (20.80%), industrial (16.84%), and commercial (9.11%) sectors.
Combustion of these fuels in traditional technologies such as Fixed Chimney Bulls Trench Kiln
(FCBTK) and low efficiency engines (vehicles, captive power generator sets etc.) emit
significant amounts of greenhouse gases and air pollutants. This has contributed to elevated
ambient concentrations of particulate matter (PM), including black carbon and organic carbon,
and several gaseous species such as ozone, polycyclic aromatic hydrocarbons (PAHs),
acetonitrile, benzene and isocyanic acid (Pudasainee et al., 2006; Aryal et al., 2009; Panday and
Prinn, 2009; Sharma et al., 2012; World Bank, 2014; Chen et al., 2015; Putero et al., 2015: Sarkar
et al., 2016). The ambient levels often exceed national air quality guidelines (Pudasainee et al.,
2006; Aryal et al., 2009; Putero et al., 2015) and are comparable or higher than ambient levels
observed in other major cities in South Asia.
Past studies in the Kathmandu Valley have focused mainly on a few aerosols species (BC, PM)
and short-lived gaseous pollutants such as ozone and carbon monoxide (Pudasainee et al., 2006;
Aryal et al., 2009; Panday and Prinn, 2009; Sharma et al., 2012, Putero et al., 2015). To the best
of authors' knowledge, no direct measurements of $CO_2$ and $CH_4$ are available for the Kathmandu
Valley. Recently, emission estimates of $CO_2$ and $CH_4$ were derived for the Kathmandu Valley
using the International Vehicle Emission (IVE) model (Shrestha et al., 2013). The study
estimated 1554 Gg of annual emission of $CO_2$ from a fleet of vehicles (that consisted of public
buses, 3-wheelers, taxis and motor cycles; private cars, trucks and non-road vehicles were not
included in the study) for the year 2010. In addition, the study also estimated 1.261 Gg of $CH_4$
emitted from 3 wheelers (10.6 %), taxis (17.7 %) and motorcycles (71 %) for 2010.
This study presents the first 12 months of measurements of two key GHGs, $CH_4$ and $CO_2$ along
with other trace gases and meteorological parameters in Bode, a semi-urban site in the eastern
part of the Kathmandu Valley. The year-long measurement in Bode is a part of the SusKat-ABC
(Sustainable Atmosphere for the Kathmandu Valley – Atmospheric Brown Clouds) international
air pollution measurement campaign conducted in and around the Kathmandu Valley from
December 2012 to June 2013. Details of the SusKat-ABC campaign are described in Rupakheti
et al. (2016, manuscript in preparation). The present study provides a detailed account of



seasonal and diurnal behaviors of $CO_2$ and $CH_4$ and their possible sources. To examine the rural-
urban differences and estimate the urban enhancement, these gaseous species were also
simultaneously measured for about three months (Jul-Oct) in 2015 at Chanban, a rural site about
25 km (aerial distance) outside and southwest of Kathmandu Valley. The seasonality of the trace
gases and influence of potential sources in various (wind) directions are further explored by via
ratio analysis. This measurement provides unique data from highly polluted but relatively poorly
studied region (central Himalayan foothills in South Asia) which could be useful for validation
of emissions estimates, model outputs and satellite observations. The study, which provides new
insights on potential sources, can also be a good basis for designing mitigation measures for
reducing emissions of air pollutants and controlling greenhouse gases in the Kathmandu Valley
and the region.
**2   Experiment and Methodology**
**2.1 Kathmandu Valley**
The Kathmandu Valley consists of three administrative districts: Kathmandu, Lalitpur, and
Bhaktapur, situated between 27.625° N, 27.75° N and 85.25°E, 85.375°E. It is a nearly circular
bowl-shaped valley with a valley floor area of approximately 340 km$^2$ located at an altitude of
1300 m mean sea level (masl). The surrounding mountains are close to 2000-2800 in height
above sea level with five mountain passes located at about 200-600 m above the valley floor and
an outlet for the Bagmati River southwest of the Kathmandu Valley. Lack of decentralization in
in Nepal has resulted in the concentration of economic activities, health and education facilities,
the service sector, as well as most of the central governmental offices in the Kathmandu Valley.
Consequently, it is one of the fastest growing metropolitan areas in South Asia with a current
population of about 2.5 million, and the population growth rate of 4% per year (World Bank,
2013) Likewise, approximately 50% of the total vehicle fleet (2.33 million) of the country is in
Kathmandu Valley (DoTM, 2015). The consumption of fossil fuels such as liquefied petroleum
gas (LPG), kerosene for cooking and heating dominates the residential consumption, while the
rest use biofuel (fuelwood, agro-residue, animal dung) for cooking and heating in the Kathmandu
Valley. The commercial sector is also growing in the valley, and the latest data indicate the





presence of 633 industries of various sizes. These are mainly associated with dyeing, brick kilns,
and manufacturing industries. Fossil fuels such as coal and biofuels are the major fuels used in
brick kilns. Brick kilns are reported as one of the major contributors of air pollution in the
Kathmandu Valley (Chen et al., 2015; Kim et al., 2015; Sarkar et al., 2016). There are about 115
brick industries in the valley (personal communication with M. Chitrakar, President of the
Federation of Nepalese Brick Industries). Acute power shortage in the Valley is common all
around the year, especially in the dry season (winter/pre-monsoon) when the power cuts can last
up to 12 hours a day (NEA, 2014). Energy demand during the power cut period is met with the
use of small (67% of 776 generators surveyed for the World Bank study was with capacity less
than 50kVA) but numerous captive power generators (diesel/petrol), which further contribute to
valley's poor air quality. According to the World Bank's estimate, over 250,000 such generator
sets are used in the Kathmandu Valley alone, producing nearly 200 MW of captive power, and
providing about 28% of the total electricity consumption of the valley (World Bank, 2014).
Apart from these sources, trash burning, which is a common practice (more prevalent in winter)
throughout the valley, is one of the major sources of air pollutants and GHGs.
Climatologically, Kathmandu Valley has a sub-tropical climate with annual mean temperature of
18°C, and annual average rainfall of 1400 mm, of which 90% occurs in monsoon season (June-
September). The rest of the year is dry with some sporadic rain events. The wind circulation at
large scale in the region is governed by the Asian monsoon circulation and hence the seasons are
also classified based on such large scale circulations and precipitation: Pre-Monsoon (March-
May), Monsoon (June-September), Post-Monsoon (October-November) and Winter (December-
February). Sharma et al. (2012) used the same classification of seasons while explaining the
seasonal variation of BC concentrations observed in the Kathmandu Valley. Locally in the
valley, the mountain-valley wind circulations play an important role in influencing air quality.
The wind speed at the valley floor is calm ($\leq 1$ m s$^{-1}$) in the morning and night, while a westerly
wind develops after 11:00 AM in the morning till dusk, and switches to a mild easterly at night
(Panday and Prinn, 2009; Regmi et al., 2003). This is highly conducive to building up of air
pollution in the valley, which gets worse during the dry season.
**2.2 Study sites**





Two sites, a semi-urban site within the Kathmandu Valley and a rural site outside the Kathmandu
Valley, were selected for this study. The details of the measurements carried out in these sites is
described Table 1 and in section 2.2.1 and 2.2.2.
**2.2.1 Bode (SusKat-ABC supersite)**
The SusKat-ABC supersite was set up at Bode, a semi-urban location (Figure 1) of the
Madhyapur Thimi municipality in the Bhaktapur district in the eastern side of the Kathmandu
Valley. The site is located at $27.68^0$N latitude, $85.38^0$E longitude, and 1344 masl. The local area
around the site has a number of scattered houses and agricultural fields. The agriculture fields are
used for growing rice paddies in the monsoon season. It also receives outflow of polluted air
from three major cities in the valley: Kathmandu Metropolitan City and Lalitpur Sub-
metropolitan City, both mainly during daytime, and Bhaktapur Sub-metropolitan City mainly
during nighttime. Among other local sources around the site, about 10 brick kilns are located in
the east and southeast direction, approximately within 1-4 km from the site which are operational
only during dry season (January to April). There are close to 20 small and medium industries
(pharmaceuticals, plastics, electronics, tin, wood, aluminum, iron, and fabrics etc.) scattered in
the same direction. The Tribhuvan International Airport (TIA) is located approximately 4 km
away to the west of the Bode site.
**2.2.2 Chanban**
Chanban is a rural/background site in Makwanpur district outside of the Kathmandu Valley
(Figure 1). This site is located ~25 km aerial distance due southwest from Bode. The site is
located on a small ridge ($27.65^0$N, $85.14^0$E, 1896 masl) between two villages - Chitlang and
Bajrabarahi - within the forested watershed area of Kulekhani Reservoir, which is located ~ 4.5
km southwest of the site. The instruments were set up on the roof of 1-storey building in an open
space inside the Nepali Army barrack. There was a kitchen of the army barrack at about 100 m to
the southeast of the measurement site. The kitchen uses LPG, electricity, kerosene, and firewood
for cooking activities.
**2.3 Instrumentation**





The measurements were carried out in two phases in 2013-2014 and 2015. In phase one, a cavity
ring down spectrometer (Picarro G2401, USA) was deployed in Bode to measure ambient $CO_2$,
$CH_4$, CO, and water vapor mixing ratios. Twelve months (6 March 2013 - 5 March 2014) of
continuous measurements were made in Bode. The operational details of the instruments
deployed in Bode are also provided in Table 1. In phase two, simultaneous measurements were
made in Bode and Chanban for a little less than 3 months (15 July to 03 October 2015).
The Picarro G2401 analyzer quantifies spectral features of gas phase molecules by using a novel
wavelength-scanned cavity ring down spectroscopic technique (CRDS). The instrument has a 30
km path length in a compact cavity that results high precision and sensitivity. Because of the
high precision wavelength monitor, it uses absolute spectral position and maintains accurate peak
quantification. Further, it only monitors the special features of interest for reducing the drift. The
instrument also has water correction to report dry gas fraction. The reported measurement
precision for $CO_2$, $CH_4$, CO and water vapor in dry gas is < 150 ppb, < 30 ppb, < 1ppb and < 200
ppm for 5 seconds with 1 standard deviation (Picarro, 2015).
In Bode, the Picarro analyzer was placed on the $4^{th}$ floor of a 5-storey building with an inlet at
0.5 m above the roof of the building with a 360 degree view (total inlet height: 20 m above
ground). The sample air was filtered at the inlet to keep dust and insects out and was drawn into
the instrument through a 9 m Teflon tube (1/4 inches ID). The Picarro analyzer was set to record
data in every 5 second and recorded both directly sampled data and water corrected data of $CO_2$
and $CH_4$. In this paper, only water-corrected or dry mixing ratios of $CH_4$ and $CO_2$ were used to
calculate the hourly averages for diurnal and seasonal analysis.
The instruments were factory calibrated before commencing the field measurement. Picarro
G2401 model is designed for remote application and long term deployment with minimal drift
and less requirement for intensive calibration (Crosson, 2008) and thus was chosen for the
current study in places like Kathmandu where there is no or limited availability of high quality
reference gases. Regular calibration of Picarro G2401 in field during 2013-2014 deployment was
not conducted due to challenges associated with the quality of the reference gas, especially for
CO and $CH_4$. One time calibration was performed for $CO_2$ (at 395, and 895 ppmv) in July 2015





before commencing the simultaneous measurement in Bode and Chanban in 2015. The %
difference of the analyzer differed by approximately 5% at reference mixing ratio. CO
observations from Picarro G2401 were compared with observations from another CO analyzer
(Horiba, model AP370) that was also operated in Bode for 3 months (March - May 2013).
Horiba CO monitor was a new unit, which was factory calibrated before its first deployment in
Bode. Nevertheless, this instrument was inter-compared with another CO analyzer (same model)
from the same manufacturer prior to the campaign and its correlation coefficient was 0.9 [slope
of data from the new unit (y-axis) vs the old unit (x-axis) = 1.09]. Primary gas cylinders from
Linde UK (1150 ppbv) and secondary gases from Ultra-Pure Gases and Chemotron Science
Laboratories (1790 ppbv) were used for the calibration of CO instrument. Further details on CO
measurements and calibration of Horiba AP370 can be found in Sarangi et al. (2014; 2016).
Statistically significant correlation (r = 0.99, slope = 0.96) was found between Picarro and
Horiba hourly average CO mixing ratio data (Supplementary Information Figure S1).
Furthermore, the monthly mean difference between these two instruments (Horiba AP370 minus
Picarro G2401) was calculated to be 0.02 ppm (3%), 0.04 ppm (5%) and 0.02 ppm (4%) in
March, April and May, respectively. For the comparison period of 3 months, the mean difference
was 0.02 ppm (4%). Overall differences were small to negligible during the comparison period
and thus, adjustment in the data was deemed not necessary.
Besides highly selective to individual species, Picarro G2401 has a water correction function and
thus accounts for the any likely drift in CO, $CO_2$ and $CH_4$ mixing ratios with the fluctuating
water vapor concentration (Chen et al., 2013;Crosson, 2008). Crosson (2008) also estimated a
peak to peak drift of 0.25 ppmv. Further, Crosson (2008) observed a 1.2 ppbv/day drift in $CO_2$
after 170 days from the initial calibration. For a duration of one year the drift will be less than 1
ppmv, which is less than 1% of the observed mixing ratio in (hourly ranges: 376-537 ppm) Bode
even if the drift was in same magnitude as in case of Crosson (2008). Crosson (2008) reported
0.8 ppbv peak to peak drift in $CH_4$ measurements for 18 days after the initial calibration.
There were other instruments concurrently operated in Bode; a ceilometer for measuring mixing
layer height (Vaisala Ceilometer CL31, Finland), and an Automatic Weather Station (AWS)
(Campbell Scientific, USA). The ceilometer was installed on the rooftop (20 m above ground) of





the building (Mues et al., 2017). For measuring the meteorological parameters, a Campbell Scientific AWS (USA) was set up on the roof of the building with sensors mounted at 2.9 m above the surface of the roof (22.9 m from the ground). The Campbell Scientific AWS measured wind speed and direction, temperature, relative humidity and solar radiation every minute. Temperature and rainfall data were taken from an AWS operated by the Department of Hydrology and Meteorology (DHM), Nepal at the Tribhuvan International Airport (TIA, see Figure 1), ~4 km due west of Bode site.

At Chanban, the inlet for Picarro gas analyzer was kept on the rooftop ~3 m above the ground and the sample air was drawn through a 3 m long Teflon tube (1/4 inches ID). The sample was filtered at the inlet with a filter (5-6 µm pore size) to prevent aerosol particles from getting into the analyzer. An automatic weather station (Davis Vantage Pro2, USA) was also set up in an open area, about 17 m away from the building and with the sensors mounted at 2 m above ground.

## 3. Results and discussion

The results and discussions are organized as follow: Sub-section 3.1 describes a year round variation in $CH_4$, $CO_2$, CO and water vapor at Bode; sub-sections 3.2, 3.3 and 3.4 present the analysis of the observed diurnal, monthly, seasonal variations. Sub-sections 3.5, 3.6, 3.7 discusses the impact of city pollution at the measurement site at Bode, influence of regional pollution and potential sources in the valley and sub-section 3.8 compares and contrasts $CH_4$, $CO_2$, CO at Bode and Chanban.

### 3.1 Time series of $CH_4$, $CO_2$, CO and water vapor mixing ratios

Figure 2 shows the time series of hourly mixing ratios of $CH_4$, $CO_2$, CO, and water vapor at Bode. Meteorological data from Bode and the Tribhuvan International Airport are also shown in Figure 2. Data gaps in Figure 2a and 2b were due to maintenance of the measurement station. In general, the changes observed in CO mixing ratio was higher in terms of % change than the variations observed in $CH_4$ and $CO_2$ mixing ratios during the sampling period. In contrast, CO mixing ratios decreased and water vapor mixing ratios increased significantly during the rainy





season (June-September). For the entire sampling period, the annual average $CH_4$, $CO_2$, CO, and
water vapor mixing ratios were 2.193 (± 0.224) ppm, 419.4 (± 23.9) ppm, 0.50 (± 0.35) ppm, and
1.71 (± 0.71) %, respectively. The annual $CH_4$ and $CO_2$ mixing ratios were compared to the
historical background site (Mauna Loa Observatory, Hawaii, USA) and the background site
(Waliguan, China) in Asia, which will provide insight on spatial differences. The selection of
neighboring countries' (i.e., India and China's) urban and semi-urban sites, where many
emission sources are typical for the region, for comparison provides information on relative
differences (higher/lower), which will help in investigating possible local emission sources in the
valley. As expected, annual mean of $CH_4$ and $CO_2$ mixing ratios in the Kathmandu Valley were
higher than the levels observed at background sites in the region and elsewhere for the same
period (Table 4). $CH_4$ was nearly 20% higher at Bode than at Mauna Loa observatory (1.831
ppm) (Dlugokencky et al., 2016) and 17% higher than at Mt. Waliguan (1.879 ppm) in China for
the same observation period (Dlugokencky et al., 2016). The small difference between Bode and
Waliguan in comparison to Mauna Loa Observatory indicates the higher mixing ratio of $CH_4$ in
Asia region. It could be associated with agricultural activities in this region. Similarly, the annual
$CH_4$ at Bode was higher than urban/semi-urban sites in India, such as an urban site in
Ahmedabad (1.880 ppm) (Sahu and Lal, 2006) and Shadnagar (1.92 ± 0.07), a semi-urban site in
Telangana state (~70 km north from Hyderabad city) during 2014 (Sreenivas et al., 2016).
Likewise, the annual average $CO_2$ mixing ratio at Bode (419.4 ppm) during the observation
period was 5.7% higher than at Mauna Loa Observatory (396.76 ppm) (Tans and Keeling, 2014)
and 5.5% higher than at Mt. Waliguan (397.7 ppm). The $CO_2$ mixing ratio in the Kathmandu
Valley was also found to be higher than the levels observed in Shadnagar (394 ± 2.9 ppm) during
2014, Ahmedabad city (413 ± 13.7 ppm) in India during November 2013 to May 2015, and an
urban site at Nanjing (406.5 ± 20 ppm) in China (Huang et. al., 2015; Sreenivas et al., 2016;
Chandra et al., 2016).

The high $CH_4$ and $CO_2$ mixing ratios at Bode in comparison to Ahmedabad, Shadnagar and
Nanjing could be due to more than 115 coal-biomass fired brick kiln, some of them are located
near the site (less than 4 km) and confinement of pollutants within the Valley due to bowl shaped
topography of the Kathmandu Valley. Although Ahmedabad and Nanjing sites are in big cities



with high population larger than Kathmandu Valley but they are far from the nearby heavy
polluting industries and situated in plains, where ventilation of pollutants would be more
efficient as opposed to the Kathmandu Valley. The major polluting sources were industries,
residential cooking and transport sector in Ahmedabad (Chandra et al., 2016). Anthropogenic
emission, synoptic circulation, terrestrial biosphere had important role on $CO_2$ mixing ratios in
Nanjing (Huang et al., 2015). Shadnagar is a small town with a population of 0.16 million and
major sources were industries (small-medium), biomass burning in residential cooking
(Sreenivas et al., 2016).
The monthly average of $CO_2$ mixing ratios in 2015 in Chanban (Aug: 403.4, Sep: 399.1 ppm)
were slightly higher than the background sites at Mauna Loa Observatory (Aug: 398.89 ppm,
Sep: 397.63 ppm) and Mt. Waliguan (Aug: 394.55 ppm, Sep: 397.68 ppm) (Dlugokencky et al.,
2016). For these two months in 2015, $CH_4$ mixing ratios were also higher in Bode (Aug: 2281.11
ppb, Sep: 2370.93 ppb) and Chanban (Aug: 2049.71 ppb, Sep: 2101.75 ppb) compared to Mauna
Loa Observatory (Aug: 1831.04 ppb, Sep: 1845.68 ppb) (Dlugokencky et al., 2016)) and Mt.
Waliguan (Aug: 1914.99 ppb, 1911.21 ppb) (Dlugokencky et al., 2016). The low differences in
$CO_2$ between Chanban and background sites mentioned above indicate the less number of and/or
less intense $CO_2$ sources at Chanban during these months because of the lack of burning
activities due to rainfall in the region. However, high $CH_4$ values in August and September in
Bode, Chanban and Mt. Waliguan in comparison to Mauna Loa Observatory may indicate the
influence of $CH_4$ emission from paddy fields in the Asian region.
**3.2 Monthly and Seasonal variations**
Figure 3 shows the monthly box plot of hourly $CH_4$, $CO_2$, CO and water vapor observed for a
year in Bode. Monthly and seasonal averages of $CH_4$ and $CO_2$ mixing ratios at Bode are
summarized in Table 2 and 3. $CH_4$ were lowest during May-July (ranges from 2.093-2.129 ppm)
period and highest during August-September (2.274-2.301 ppm), followed by winter. In addition
to the influence of active local sources, the shallow boundary layer in winter was linked to
elevated concentrations (Panday and Prinn, 2009; Putero et al., 2015, Mues et al., 2016). The low
$CH_4$ values from May to July may be associated with the absence of brick kiln and frequent





rainfall in these months. Brick kiln were operational during January to April. Rainfall also leads
to suppression of open burning activities in the valley (see Figure 2b). The $CH_4$ was slightly
higher (statistically significant, $p<0.05$) in monsoon season (July –September) than pre-monsoon
season (unlike $CO_2$ which was higher in pre-monsoon), and could be associated with the addition
of $CH_4$ flux from the water-logged rice paddies (Goroshi et al. (2011). There was a visible drop
in $CH_4$ from September to October but remained consistently over 2.183 ppm from October to
April with little variation between these months. Rice-growing activities are minimal or none in
October and beyond, and thus may be related to the observed dip in $CH_4$ mixing ratio.
Comparison of seasonal average $CH_4$ mixing ratios at Bode and Shadnagar (a semi-urban site in
India) indicated that $CH_4$ mixing ratios at Bode were higher in all seasons than at Shadnagar:
pre-monsoon ($1.89 \pm 0.05$ ppm), monsoon ($1.85 \pm 0.03$ ppm), post-monsoon ($2.02 \pm 0.01$ ppm),
and winter ($1.93 \pm 0.05$ ppm) (Sreenivas et al., 2016). The possible reason for lower $CH_4$ at
Shadnagar in all season could be associated with geographical location and difference in local
emission sources. The highest $CH_4$ mixing ratio in Shadnagar was reported in post-monsoon
which was associated with harvesting in the Kharif season (July – October), while the minimum
was in monsoon. Shadnagar is a relatively small city (population: ~0.16 million) compared to
Kathmandu Valley and the major local sources which may have influence on $CH_4$ emission
include bio-fuel, agro-residue burning and residential cooking.
The seasonal variation in $CO_2$ generally reflects the seasonality of major emission sources such
as brick kilns and regional emission sources such as vegetation fire and agriculture residue burn.
The concentrations of most pollutants in the region are lower during the monsoon period
(Sharma et al., 2012, Marinoni, 2013; Putero et al., 2015) due to limited emission sources and
partially due to rain washout. Monsoon is also the growing season with higher $CO_2$ assimilation
by plants than other seasons (Sreenivas et al., 2016). In contrast, winter, pre-monsoon and post-
monsoon season experiences an increase in emission activities in the Kathmandu Valley (Putero
et al., 2015).
The $CO_2$ mixing ratios were in the range of 376 - 537 ppm for the entire observation period.
Differences with $CH_4$ were observed in September and October where $CO_2$ was increasing




(mean/median) in contrast to $CH_4$ which showed the opposite trend. The observed increase in
$CO_2$ after October may be related to less or no rainfall, which results in the absence of rain-
washout and/or no suppression of active emission sources such as open burning activities. $CO_2$
remains relatively lower during July-August, but it is over 420 ppm from January to May.
Seasonal variation of $CO_2$ in Bode was similar in seasonal variation but the values are higher
than the values observed in Shadnagar, India (Sreenivas et al., 2016).
The variations in CO were more distinct than $CH_4$ and $CO_2$ during the observation period
(Figure 3). The highest CO values were observed from January-April (0.71-0.91 ppm). The
seasonal mean of CO mixing ratios at Bode were: pre-monsoon (0.60 ±0.36 ppm), monsoon
(0.26±0.09 ppm), post-monsoon (0.40±0.15 ppm), and winter (0.76±0.43 ppm). The maximum
CO was observed in winter, unlike $CO_2$ which was maximum in pre-monsoon. The high CO in
winter was due to the presence of strong local pollution sources (Putero et al., 2015) and shallow
mixing layer heights. The addition of regional forest-fire and agro-residue burning augmented
$CO_2$ mixing ratios in pre-monsoon. The water vapor mixing ratio showed a seasonal pattern
opposite of CO, with a maximum in monsoon (2.53 %) and minimum in winter (0.95 %), and
intermediate values of 1.56 % in pre-monsoon and 1.55 % in post-monsoon season.
There were days in August-September when the $CH_4$ increases by more than 3 ppm (Figure 2).
Enhancement in $CO_2$ was also observed during the same time period. It is likely that these high
enhancements were associated with the air mass from Northeast-East (NE-E) which had > 2.5
ppm $CH_4$ and > 450 $CO_2$ (see Figure 4). CO during the same period was not enhanced and didn't
show any particular directionality compared to $CH_4$ and $CO_2$ (not shown in Figure 4). Areas NE-
E to Bode are predominantly irrigated (rice paddies) during August-September, and sources such
as brick kilns were not operational during this time period. Goroshi et al. (2011) reported that
June to September is a growing season for rice paddies in South Asia with high $CH_4$ emissions
during these months and observed a peak in September in the atmospheric $CH_4$ column over
India. Model analysis also points to high methane emissions in September which coincides with
the growing period of rice paddies (Goroshi et al., 2011, Prasad et al., 2014). The $CH_4$ mixing
ratios at Bode in January (2.233 ± 0.219 ppm) and July (2.129 ± 0.168 ppm) were slightly higher
than the observation in Darjeeling (Jan: 1.929±0.056 ppm; Jul: 1.924±0.065 ppm), a hill station



of eastern Himalaya (Ganesan et al., 2013). The higher $CH_4$ values in January and July at Bode
compared to Darjeeling could be because of the influence of local sources, in addition to the
shallow boundary layer in Kathmandu Valley. Trash burning and brick kilns are two major
sources from December until April in the Kathmandu Valley while emission from paddy fields
occurs during July-September in the Kathmandu Valley. In contrast, the measurement site in
Darjeeling was located at higher altitude (2194 masl) and was less influenced by the local
emission. The measurement in Darjeeling reflected a regional contribution. There are limited
local source in Darjeeling such as wood biomass burning, natural gas related emission and
vehicular emission (Ganesan et al., 2013).
The period between January and April had generally higher or the highest values of $CO_2$, $CH_4$
and CO at Bode. The measurement site was impacted mainly by local Westerly-Southwesterly
winds (W-SW) and East-Southeast (E-SE). The W-SW typically has a wind speed in the range
~1 - 6 m s$^{-1}$ and was active during late morning to afternoon period (~11:00 to 17:00 NST,
supplementary information Figure S2 and S3). Major cities in the valley such as Kathmandu
Metropolitan City and Lalitpur Sub-metropolitan City are W-SW of Bode (Figure 1c). Wind
from E-SE were generally calm ($\leq$1m s$^{-1}$) and observed only during night and early morning
hours (21:00 to 8:00 NST). The mixing ratio of all three species in air mass from the E-SE was
significantly higher than in the air mass from W-SW (Figure 4). There are 10 biomass co-fired
brick kilns and Bhaktapur Industrial Estate located within 1-4 km E-SE from Bode (Sarkar et al.,
2016). The brick kilns were only operational during January-April. Moreover, there were over
100 brick kilns operational in the Kathmandu Valley (Putero et al., 2015) which use low-grade
lignite coal imported from India and biomass fuel to fire bricks in inefficient kilns (Brun, 2013).
Fresh emissions from main city center were transported to Bode during daytime by W-SW winds
which mainly include vehicular emission. Compared to monsoon months (June-August), air
mass from W-SW had higher values in all three species (Figure 4) during winter and pre-
monsoon months. This may imply that in addition to vehicular emission, there are other potential
sources which were exclusively active during these dry months.  Municipal trash burning is also
common in the Kathmandu Valley, with a reported higher frequency from December to February
(Putero et al., 2015). The frequency in the use of captive power generator sets are highest during



the same period, which is another potential source contributing to air coming from W-SW
direction (World Bank, 2014; Putero et al., 2015).
Regional transport of pollutants into the Kathmandu Valley was reported by Putero et al. (2015).
The westerly circulation (originated at longitude about $60^0$E in 5 days back trajectories) was
dominant from March-May 2013. Other sources of $CO_2$ and $CH_4$ could be due to vegetation fires
which were also reported in the region surrounding the Kathmandu Valley during the pre-
monsoon months (Putero et al., 2015). Similarly, high pollution events, peaking in the pre-
monsoon, were observed at Nepal Climate Observatory-Pyramid (NCO-P) near Mt. Everest,
which have been associated with vegetation fires in the Himalayan foothills and northern IGP
region (Putero et al., 2014). MODIS derived forest counts (Figure 5), which also indicated high
frequency of forest fire and farm fire from February to April and also during post-monsoon
season. It is interesting that the monthly mean $CO_2$ mixing ratio was maximum in April (430 ±
27 ppm) which could be linked to the fire events. It is likely that the westerly winds (>2.5-4.5 m
$s^{-1}$) during the daytime (supplementary information Figure S2, S3) bring additional $CO_2$ from
vegetation fires and agro-residue burning in southern plains of Nepal including the IGP region
(Figure 5). Low values of $CO_2$ and $CH_4$ during June-July (Figure 3) was coincident with the
rainy season, and sources such as brick kiln emission, trash burning, captive power generators,
and regional agriculture residue burning and forest fires are weak or absent during these months.
**3.3 Diurnal Variation**
Figure 6 shows the average seasonal diurnal patterns of $CH_4$, $CO_2$, CO, and water vapor mixing
ratios observed at Bode for four seasons. All the three gas species had a distinct diurnal pattern in
all seasons, characterized by maximum values in the morning hours (peaked around 7:00-9:00),
afternoon minima around 15:00-16:00, and a gradual increase through the evening until next
morning. There was no clear evening peak in $CH_4$ and $CO_2$ mixing ratios whereas CO shows an
evening peak around 20:00. The well-defined morning and evening peaks observed in CO
mixing ratios are associated with the peaks in traffic and residential activities. The $CH_4$ and $CO_2$
showed pronounced peaks in the morning hours (7:00-9:00) in all seasons with almost the same
level of seasonal average mixing ratios. CO had a prominent morning peak in winter and pre-



monsoon season, but the peak was significantly lower in monsoon and post-monsoon. The CO
(~1-1.4 ppm) around 8:00-9:00 am in winter and pre-monsoon were nearly 3-4 times higher than
in monsoon and post-monsoon season. It appears that $CH_4$ and $CO_2$ mixing ratios were
continuously building up at night until the following morning peak in all seasons. The similar
seasonal variations in $CH_4$ and $CO_2$ across all seasons could be due to their long-lived nature, as
compared to CO, whose diurnal variations are strongly controlled by the evolution of the
boundary layer. Kumar et al. (2015) also reported morning and evening peaks and an afternoon
low in $CO_2$ mixing ratios in industrial, commercial, and residential sites in Chennai in India. The
authors also found high early morning $CO_2$ mixing ratios at all sites and attributed it to the
temperature inversion and stable atmospheric condition.
The daytime low $CH_4$ and $CO_2$ mixing ratios were due to (i) elevated mixing layer height in the
afternoon (Figure 7), (ii) development of upslope wind circulation in the valley, and (iii)
development of westerly and southwesterly winds which blows through the valley during the
daytime from around 11 am to 5 pm (supplementary information Figure S2), all of which aid in
dilution and ventilation of the pollutants out of the valley (Regmi et al., 2003; Kitada and Regmi,
2003; Panday and Prinn, 2009). In addition, the daytime $CO_2$ minimum in the summer monsoon
is also associated with high photosynthetic activities in the valley as well as in the broader
surrounding region. In the nighttime and early morning, the mixing layer height was low (only
around 200-300 m in all seasons) and stable boundary layer for almost 17 hours a day. In the
daytime it grows up to 800-1200 m for a short time (ca. from 11:00 to 6:00) (Mues et al., 2016,
manuscript submitted to ACPD). Therefore the emissions from various activities in the evening
after 18:00 (cooking and heating, vehicles, trash burning, and bricks factories in the night and
morning) were trapped within the collapsing and shallow boundary layer, and hence mixing
ratios were high during evening, night and morning hours. Furthermore, plant and soil respiration
also increases $CO_2$ mixing ratio during the night (Chandra et al., 2016). However, Ganesan et al.
(2013) found a distinct diurnal cycle of $CH_4$ mixing ratios with twin peaks in the morning (7:00-
9:00), and afternoon (15:00-17:00) and a nighttime low in winter but no significant diurnal cycle
in the summer of 2012 in Darjeeling, a hill station (2194 masl) in the eastern Himalaya. The
authors described that the morning peaks could be due to the radiative heating of the ground in



the morning, which breaks the inversion layer formed during night, and as a result, pollutants are
ventilated from the foothills up to the site. The late afternoon peaks match wind direction and
wind speed (upslope winds) that could bring pollution from plains to mountains.
The diurnal variation of CO is also presented along with $CO_2$ and $CH_4$ in Figure 6c. CO is an
indicator of primary air pollution. Although CO mixing ratio showed distinct diurnal pattern, it
was different from the diurnal patterns of $CO_2$ and $CH_4$. CO diurnal variation showed distinct
morning and evening peaks, afternoon minima, and a nighttime accumulation or decay.
Nighttime accumulation in CO was observed only in winter and pre-monsoon and decay or
decrease in monsoon season and post-monsoon season (Figure 7). The lifetime of CO (weeks to
months) is very long compared to the ventilation timescales for the valley, so the different
diurnal cycles would be due to differences in nighttime emissions. While the biosphere respires
at night, most CO sources except brick kilns remain shut down late at night. This also explains
why nighttime values of CO drop less in the winter and pre-monsoon than in other seasons.
Furthermore, the prominent morning peaks of CO in pre-monsoon and winter compared to other
seasons results from nighttime accumulation, additional fresh emissions in the morning and
recirculation of the pollutants due to downslope katabatic winds (Pandey and Prinn, 2009;
Panday et al., 2009). Pandey and Prinn (2009) observed nighttime accumulation and gradual
decay during the winter (January 2005). The measurement site in Pandey and Prinn (2009) was
near the urban core of the Kathmandu Valley and had significant influence from the vehicular
sources all over the season including the winter season. Measurement in Bode lies in close
proximity to the brick kilns which operate 24 hours during the winter and pre-monsoon period.
Calm southeasterly winds are observed during the nighttime and early morning (ca.22:00 – 8:00)
in pre-monsoon and winter, which transport emissions from brick kiln to the site (Sarkar et al.,
2016). Thus the gradual decay in CO was not observed in Bode.
The timing of the CO morning peak observed in this study matches with observations by Panday
et al. (2009). They also found CO morning peak at 8:00 in October 2004 and at 9:00 in January
2005. The difference could be linked to the boundary layer stability. As the sun rises later in
winter, the boundary layer stays stable for a longer time in winter keeping mixing ratios higher in
morning hours than in other seasons with an earlier sunrise.



The morning peaks of $CO_2$ and $CH_4$ mixing ratios occurred around 6:00-7:00 local time in the
pre-monsoon, monsoon, and post monsoon season, whereas in winter their peaks are delayed by
1-2 hours in the morning; $CH_4$ at 8:00 and $CO_2$ at 9:00. The CO showed that its morning peak
was delayed compared to $CO_2$ and $CH_4$ morning peaks by 1-2 hour in pre-monsoon, monsoon
and post-monsoon (at 8:00) and in winter (at 9:00). The occurrence of morning peaks in $CO_2$ and
$CH_4$ 1-2 hours earlier than CO is interesting. This could be due to the long lifetimes and
relatively smaller local sources of $CH_4$ and $CO_2$, as CO is mainly influenced by emissions from
vehicles during rush hour, as well as from biomass and trash burning in the morning hours. Also,
CO increases irrespective of change in mixing layer (collapsing or/rising, Figure 7) but $CO_2$ and
$CH_4$ start decreasing only after the mixing layer height starts to rise. Recently, Chandra et al.
(2016) also reported that the $CO_2$ morning peak occurred earlier than CO in observations in
Ahmedabad City India. This was attributed to $CO_2$ uptake by photosynthetic activities after
sunrise but CO kept increasing due to emissions from the rush hour activities.
Highest daytime minimum of $CO_2$ was observed in pre-monsoon which may indicate the
influence of regional emissions that increased the baseline background concentrations as well.
The daytime minimum mixing ratios occurs from 12:00 to 17:00 LST. The highest minimum
$CO_2$ was found in pre-monsoon (Figure 6b). Although the local emission sources are similar in
pre-monsoon and winter, the higher minimum daytime $CO_2$ mixing ratios in pre-monsoon season
than other seasons, suggest the influence of regional emissions in the Kathmandu Valley, which
has been reported in previous study by Putero et al. (2015). In monsoon and post-monsoon
seasons, the minimum $CO_2$ mixing ratios in the afternoon drops down to 390 ppm, which were
close to the value observed at the regional background sites such as Mauna Loa and Waliguan.
**3.4 Seasonal interrelation of $CO_2$, $CH_4$ and CO**
The Pearson's correlation coefficient (r) between $CO_2$ and CO was strong in winter (0.87),
followed by monsoon (0.64), pre-monsoon (0.52) and post-monsoon (0.32). The higher
coefficient in winter indicates that common or similar sources for $CO_2$ and CO and moderate
values in pre-monsoon and monsoon indicates the likelihood of different sources. To avoid the
influence of strong diurnal variations observed in the valley, daily averages, instead of hourly,





were used to calculate the correlation coefficients. The correlation coefficients between daily
$CH_4$ and $CO_2$ for four seasons are as follows: winter (0.80), post-monsoon (0.74), pre-monsoon
(0.70) and monsoon (0.22). A semi-urban measurement study in India also found a strong
positive correlation between $CO_2$ and $CH_4$ in the pre-monsoon (0.80), monsoon (0.61), post-
monsoon (0.72) and winter (0.8) (Sreenivas et al., 2016). It should be noted here that Sreenivas
et al., (2006) used hourly average $CO_2$ and $CH_4$ mixing ratios. The weak monsoon correlation at
Bode, which is in contrast to Sreenivas et al. (2016), may point to the influence of dominant $CH_4$
emission from paddy field during the monsoon season (Goroshi et al., 2011). Daily $CH_4$ and CO
was also weakly correlated in monsoon (0.34) and post-monsoon (0.45). Similar to $CH_4$ and
$CO_2$, the correlation between $CH_4$ and CO were moderate to strong in pre-monsoon (0.76) and
winter (0.75).
Overall, the positive and high correlations between $CH_4$ and CO mixing ratios and between $CH_4$
and $CO_2$ in pre-monsoon and winter indicate common sources or source regions, most likely
combustion related sources such as vehicular emission, brick kilns, agriculture fire etc. Weak
correlation, between $CH_4$-$CO_2$ and between $CH_4$-CO, during monsoon season indicates sources
other than combustion-related may be active, such as agriculture as a key $CH_4$ source (Goroshi et
al., 2013)
**3.5 Influence of regional emission and transport**
Regional sources and transport can influence the level of air pollution in the Kathmandu Valley
mainly originating from regions west of the Kathmandu Valley (Putero et al., 2015). Wind from
the north, which is less frequent than southerly and westerly winds, often brings cleaner air mass
(also low in $CH_4$, Figure 4) and hence helps dilute or flush out the valley's polluted air.
Household combustion of biofuel, used mainly in the southern plains of Nepal and the IGP
region, is an important contributor to the regional pollution in the higher mountainous areas
(Panday and Prinn, 2009; Putero et al., 2014). Recently, Putero et al. (2015) attributed the
afternoon high BC and $O_3$ concentrations at Paknajol in the Kathmandu Valley during pre-
monsoon season to regional vegetation fire episodes and linked to the regional transport by
westerly circulation. Our study also observed a number of episodes with high $CO_2$, $CH_4$ and CO





mixing ratios at Bode during most of the days in March, April and May. During the entire
sampling period of a year, there were 42 days with $CO_2$ mixing ratio $\geq$ 430 ppm, of which 29
days (or 69%) were during the pre-monsoon (25 days or 59% in March and –April alone) and 10
days (23%) in winter. However, atmospheric chemistry transport models are required to confirm
and differentiate contributions of local sources and regional sources influencing the Kathmandu
Valley, which is beyond the scope of this study.

**3.6 CO and $CO_2$ ratio: Potential emission sources**
The ratio of the ambient mixing ratios of CO and $CO_2$ was used as an indicator to help
discriminate emission sources in the Kathmandu Valley. The ratio was calculated from the
excess (dCO and $dCO_2$) relative to the background values of ambient CO and $CO_2$ mixing ratios.
The excess value was estimated by subtracting the base value which was calculated as the fifth
percentile of the hourly data for a day (Chandra et al., 2016).
Average emission ratios from the literature are shown in Table 5, and average ratios of
$dCO/dCO_2$ are shown in Table 6, disaggregated into morning hours, evening hours, and seasonal
values. Higher ratios were found in pre-monsoon (12.4) and winter (15.1) season compared to
post-monsoon (8.3) and monsoon (7.5). These seasonal differences in the $dCO/dCO_2$ ratio are
depicted in Figure 8, which shows a clear relationship with the wind direction and associated
emissions, with the highest values especially for stronger westerly winds. Compared to the other
three seasons, the ratio in winter was also relatively high for air masses from the east, likely due
to emissions from brick kilns combined with accumulation during more stagnant meteorological
conditions (supplementary information Figure S2, S3).  In other seasons, emission emanating
from the north and east of Bode were characterized by a $dCO/dCO_2$ ratio below 15. Air masses
from the west and south generally have a ratio from 20 to 50 in all but post-monsoon season,
where the ratio sometimes exceeds 50. A ratio of 50 or over is normally due to very inefficient
combustion sources (Westerdahl et al., 2009; Stockwell et al., 2016), such as agro-residue
burning, which is common during the post-monsoon season in the Kathmandu Valley.





For interpretability of emission ratio with sources, the ratio was classified into three categories:
(i) 0 – 15, (ii) 15 – 45, and (iii) greater than 45. This classification was based on the observed
distribution of emission ratio during the study period (Figure 8) and a compilation of observed
emission ratios typical for different sources from Nepal and India (see Table 5). An emission
ratio below 15 is likely to indicate residential cooking and diesel vehicles, and captive power
generation with diesel-powered generator sets (Smith et al., 2000; ARAI, 2008; World Bank,
2014). The emission from brick kilns (FCBTK and Clamp kilns, both common in the Kathmandu
Valley), and inefficient, older (built before 2000) gasoline cars fall in between 15 - 45 (Weyant
et al., 2014, Stockwell et al., 2016; ARAI, 2008). Four-stroke motorbikes and biomass burning
activities (mixed garbage, crop-residue and biomass) are one of the least efficient combustion
sources, with emission ratios higher than 45 (Westerdahl et al., 2009; Stockwell et al., 2016;
ARAI, 2008).
Based on the classification and Figure 8, the emissions from sources to the north and east of the
site are dominated by residential cooking and/or diesel combustion. Emissions from the south
and west of Bode are mainly contributed by sources such as brick kilns and inefficient gasoline
vehicles. Very high ratios, indicative of agro-residue open burning, generally only show up
during the post-monsoon period, when such activities take place, especially in areas southwest of
the site. The relatively enhanced ratio (20-30) observed in winds from north and east of the site
during winter is mostly likely due to brick kilns that use mixed coal-biomass fuel, whereas the
Figure 8 indicates the dominant signature of residential cooking, diesel and old gasoline cars
during the pre-monsoon, monsoon and post-monsoon seasons.
The dCO/dCO$_2$ ratio also changes markedly between the morning peak hours (7:00-9:00, except
in winter season when the peak occurs during 8:00-9:00) and evening peak hours (19:00-21:00
pm) (Table 6). Morning and evening values were lowest (2.2, 8.0) during the monsoon and
highest (11.2, 21.6) in the winter season, which points to the different emission characteristics in
these two seasons. This feature is similar to Ahmedabad, India, another urban site in south Asia,
where the morning/evening values were lowest (0.9/19.5) in monsoon and highest in winter
(14.3/47.2) (Chandra et al., 2016). In the morning period, the ratio generally falls within a
narrower range, from less than 1 to about 25, which indicates a few dominant sources, such as



cooking, diesel vehicles, and diesel gen-sets (see Figure 9). In the evening period, the range of the ratio is much wider, from less than 1 to more than 100, especially in winter. This is partly due to the shallower boundary layer in winter, giving local CO emissions a chance to build up more rapidly compared to the longer-lived and well-mixed $CO_2$, and also indicating the prevalence of additional sources such as brick kilns and agro-residue burning.

### 3.7 Comparison of $CH_4$ and $CO_2$ at semi-urban site (Bode) and rural site (Chanban)

Figure 10 shows time series of hourly average mixing ratios of $CH_4$, $CO_2$, CO and water vapor observed simultaneously at Bode and Chanban for the period of 15[th] July to 3[rd] October 2015. The hourly meteorological parameters observed at Chanban are shown in supplementary Figure S4. The hourly temperature ranges from 14 to 28.5 °C during the observation period. The site experienced calm winds during the night and moderate southeasterly winds with hourly maximum speed of up to 7.5 m s[-1] during the observation period. The $CH_4$ mixing ratios at Chanban varied from 1.880 ppm to 2.384 ppm, and generally increased from the last week of July until early September, peaking around 11[th] September and then falling off towards the end of the month. CO followed a generally similar pattern, with daily average values ranging from 0.10 ppm to 0.28 ppm. The hourly $CO_2$ mixing ratios ranged from 375 to 453 ppm, with day to day variations, but there were no clear pattern as observed in trend like $CH_4$ and CO mixing ratios.

The $CH_4$, $CO_2$, and CO mixing ratios were higher in Bode than in Chanban (Figure 10, Table 4), with Chanban approximately representing the baseline of the lower envelope of the Bode levels. The mean $CO_2$, $CH_4$ and CO mixing ratios over the entire sampling period of nearly three months at Bode are 3.8%, 12.1%, and 64% higher, respectively, than at Chanban. The difference in the $CO_2$ mixing ratio could be due to the large uptake of $CO_2$ in the forested area at Chanban and surrounding regions compared to Bode, where the local anthropogenic emissions rate is higher and less vegetation for photosynthesis. The coincidence between the base values of CO and $CH_4$ mixing ratios at Bode and the levels observed at Chanban implies that Chanban CO and $CH_4$ mixing ratios are indicative of the regional background levels. A similar increase in CO and $CH_4$ mixing ratios at Chanban from July to September was also observed at Bode, which may



imply that the regional/background levels in the broader Himalayan foothill region also
influences the baseline of the daily variability of the pollutants in the Kathmandu Valley,
consistent with Panday and Prinn (2009).
Figure 11 shows the comparison of average diurnal cycles of $CO_2$, $CH_4$, CO and water vapor
mixing ratios observed at Bode and Chanban. The diurnal pattern of $CO_2$ mixing ratios at both
sites is similar, but more pronounced at Bode, with a morning peak around 6:00-7:00, a daytime
minimum, and a gradual increase in the evening until the next morning peak. A prominent
morning peak at Bode during the monsoon season indicates the influence of local emission
sources. The daytime $CO_2$ mixing ratios are also higher at Bode than at Chanban because of local
emissions less uptake of $CO_2$ for photosynthesis in the valley in comparison to the forested area
around Chanban. Like the diurnal pattern of $CO_2$ depends on the evolution of the mixing layer at
Bode, as discussed earlier, it is expected that the mixing layer evolution similarly influences the
diurnal $CO_2$ mixing ratios at Chanban. CO, on the other hand, shows very different diurnal
patterns at Bode and Chanban. Sharp morning and evening peaks of CO are seen at Bode,
indicating the strong local polluting sources, especially cooking and traffic in the morning and
evening peak hours. Chanban, in contrast, only has a subtle morning peak and no evening peak.
After the morning peak, CO sharply decreases at Bode but not at Chanban. The growth of the
boundary layer after sunrise and entrainment of air from the free troposphere, with lower CO
mixing ratios, causes CO to decrease sharply during the day at Bode. At Chanban, on the other
hand, since the mixing ratios are already more representative of the local and regional
background levels which will also be prevalent in the lower free troposphere, CO does not
decrease notably during the daytime growth of the boundary layer as observed at Bode.
Similarly, while there is very little diurnal variation in the $CH_4$ mixing ratios at Chanban, there is
a strong diurnal cycle of $CH_4$ at Bode, similar to $CO_2$ there. At Chanban, the $CH_4$ mixing ratio
only shows a weak minimum at around 11 am, a slow increase during the day until a its peak
around 22:00, followed by a slow decrease during the night and a more rapid decrease through
the morning. The cause of this diurnal pattern at Chanban is presently unclear, but it is clear that
the levels are generally representative of the regional background throughout the day and show
only limited influences of local emissions.



## 4. **Conclusions**

A cavity ring down spectrometer (Picarro G2401, USA) was used to measure ambient $CO_2$, $CH_4$, CO, and water vapor mixing ratios at a semi-urban site (Bode) in the Kathmandu Valley for a year. This was the first 12-months of continuous measurements of these four species in the Kathmandu Valley in the foothills of the central Himalaya. Simultaneous measurement was carried out at a rural site (Chanban) for approximately 3 months to evaluate urban-rural differences.

The measurement also provided an opportunity to establish diurnal and seasonal variation of these species in one of the biggest metropolitan cities in the foothills of Himalayas. Annual average of the mixing ratio of $CH_4$ and $CO_2$ in Bode revealed that they were higher than the concentrations at the background sites such as the Mauna Loa, USA and Mt. Waliguan, China, as well as higher than urban/semi-urban sites in nearby regions such as Ahmedabad and Shadnagar in India, and Nanjing in China. These comparisons highlight potential sources of $CH_4$ and $CO_2$ in the Kathmandu Valley, such as brick kilns in the valley.

Polluted air masses were transported to the site mainly by two major local wind circulation patterns, East-South/North East and West-Southwest throughout the observation period. Strong seasonality was observed with CO compared to $CO_2$ and $CH_4$. Winter and pre-monsoon high CO are linked to emission sources active in these seasons only and are from east-southeast and west-southwest. Emission from the east-southeast are most likely related to brick kilns (winter and pre-monsoon), which are in close proximity to Bode. Major city-centers are located in the west-southwest of Bode (vehicular emission) which impact the site all-round the year, although higher during winter season. Winter high was also observed with $CO_2$ and $CH_4$, which are mostly local influence of brick kilns, trash burning and emission from city-center. Nighttime and early morning accumulation of pollutants in winter due to a shallow stable mixing height (200 m) also contribute to elevated levels than other seasons. Regional transport into the Kathmandu Valley could be related to $CO_2$ peak during pre-monsoon. The highest $CH_4$ during the post-monsoon could be associated with agricultural activity northeast of Bode. Diurnal variation across all seasons indicates the influence of rush-hour emissions related to vehicles and residential



emissions. The evolution of the mixing layer height (200-1200 m) was a major factor which
controls the morning-evening peak, afternoon low and night-early morning accumulation or
decay. Thus the geographical setting of the Kathmandu Valley and its associated meteorology
play a key role in the dispersion and ventilation of pollutants in the Kathmandu Valley. The ratio
of $CO/CO_2$ across different season and wind direction showed that emissions from inefficient
gasoline vehicles, brick kilns, residential cooking and diesel combustion are likely to impact
Bode.
The differences in mean values for urban-rural measurements at Bode and Chanban is highest for
CO (64 %) compared to $CO_2$ (3.8%) and $CH_4$ (12%). Low values of $CH_4$ and $CO_2$ mixing ratios
at the Chanban site represent a regional background mixing ratios.
This study provided valuable information on key greenhouse gases and air pollutants in the
Kathmandu Valley and the surrounding regions, useful for evaluation of satellite measurements
climate and regional air quality models. The analysis presented in the paper can provide a sound
scientific basis for reducing emissions of greenhouse gases and air pollutants in the Kathmandu
Valley.
**Acknowledgements**
The IASS is grateful for its funding from the German Federal Ministry for Education and
Research (BMBF) and the Brandenburg Ministry for Science, Research and Culture (MWFK).
This study was partially supported by core funds of ICIMOD contributed by the governments of
Afghanistan, Australia, Austria, Bangladesh, Bhutan, China, India, Myanmar, Nepal, Norway,
Pakistan, Switzerland, and the United Kingdom as well as funds provided to ICIMOD's
Atmosphere Initiative by the Governments of Sweden and Norway. We are thankful to
Bhogendra Kathayat, Shyam Newar, Dipesh Rupakheti, Ravi Pokharel, and Pratik Singdan for
their assistance during the measurement, P.S. Praveen for his support in calibration of Picarro
instrument, Pankaj Sadavarte for his help in refining Figure 1, and Liza Manandhar and Rishi
KC for the logistical support. The authors also express their appreciation to the Department of
Hydrology and Meteorology (DHM), Nepal, and the Nepal Army.



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



**Table 1.** Instruments and sampling at Bode (semi-urban site) and Chanban (rural site)

| Site | Instrument | Species | sampling interval | Measurement period | inlet/sensor height above ground (m) |
|---|---|---|---|---|---|
| Bode | i. Cavity ring down spectrometer (Picarro G2401, USA) | $CO_2$, $CH_4$, CO, water vapor | 5 sec | 06 Mar 2013 - 05 Mar 2014<br>14 Jul 2015 - 07 Aug 2015 | 20 |
| | ii. CO monitor (Horriba AP370, USA) | CO | 5 min | 06 Mar 2013 – 07 June 2013 | 20 |
| | iii. Ceilometer (Vaisala CL31, Finland | | 15-52 min | 06 Mar 2013 – 05 Mar 2014 | 15 |
| | iv. AWS (Campbell Scientific, USA) | | 1 min | | 23 |
| | a. CS215 | RH, T | | 06 Mar 2013 – 24 Apr 2013 | |
| | b. CS300 Pyranometer | SR | | 06 Mar 2013 - 05 Mar 2014<br>14 Jul 2015 - 07 Aug 2015 | |
| | c. RM Young 05103-5 | WD, WS | | 06 Mar 2013 - 05 Mar 2014<br>14 July 2015 - 07 Aug 2015 | |
| | v. Airport AWS (Environdata, Australia) | | | | |
| | a. TA10 | T | | 18 Jun 2013 – 13 Jan 2013 | |
| | b. RG series | RF | | 06 Mar 2013 – 15 Dec 2013 | |
| Chanban | i. Cavity ring down spectrometer (Picarro G2401, USA) | $CO_2$, $CH_4$, CO, water vapor | 5 sec | 15 July 2015 - 03 Oct 2015 | 3 |
| | ii AWS (Davis Vantage Pro2, USA) | RH, T, SR, WD, WS, RF, P | 10 min | 14 July 2015 - 07 Aug 2015 | 2 |

AWS: Automatic weather station, RH: ambient relative humidity, T: ambient temperature, SR: global solar radiation, WS: wind speed, WD: wind direction, RF: rainfall, P: ambient pressure



**Table 2.** Summary of monthly average $CH_4$ and $CO_2$ mixing ratios observed at Bode, a semi-urban site in the Kathmandu Valley during March 2013 to Feb 2014 [mean, standard deviation (SD), median, minimum (Min.), maximum (Max.) and number of data points of hourly average values]

| Month | CH₄ (ppm) | | | | | CO₂ (ppm) | | | | | Data points |
|---|---|---|---|---|---|---|---|---|---|---|---|
| | Mean | SD | Median | Min. | Max. | Mean | SD | Median | Min. | Max. | |
| Mar | 2.207 | 0.245 | 2.152 | 1.851 | 3.094 | 426.6 | 26.4 | 418.3 | 378.8 | 510.8 | 596 |
| Apr | 2.183 | 0.252 | 2.094 | 1.848 | 3.121 | 430.3 | 27.4 | 421.0 | 397.0 | 536.9 | 713 |
| May | 2.093 | 0.174 | 2.040 | 1.863 | 2.788 | 421.7 | 22.1 | 413.4 | 395.9 | 511.2 | 725 |
| Jun | 2.061 | 0.142 | 2.017 | 1.869 | 2.675 | 417.9 | 21.3 | 410.4 | 390.5 | 495.7 | 711 |
| Jul | 2.129 | 0.168 | 2.074 | 1.893 | 2.770 | 410.3 | 18.2 | 406.3 | 381.0 | 471.0 | 500 |
| Aug | 2.274 | 0.260 | 2.181 | 1.953 | 3.219 | 409.9 | 22.8 | 405.3 | 376.1 | 493.1 | 737 |
| Sep | 2.301 | 0.261 | 2.242 | 1.941 | 3.331 | 414.9 | 30.2 | 404.0 | 375.9 | 506.2 | 710 |
| Oct | 2.210 | 0.195 | 2.156 | 1.927 | 2.762 | 417.0 | 25.1 | 411.8 | 381.9 | 486.7 | 743 |
| Nov | 2.207 | 0.203 | 2.178 | 1.879 | 2.705 | 417.2 | 20.7 | 415.7 | 385.7 | 478.9 | 717 |
| Dec | 2.206 | 0.184 | 2.193 | 1.891 | 2.788 | 417.7 | 17.3 | 418.0 | 386.7 | 467.6 | 744 |
| Jan | 2.233 | 0.219 | 2.198 | 1.889 | 2.744 | 424.8 | 20.9 | 422.3 | 392.7 | 494.5 | 696 |
| Feb | 2.199 | 0.223 | 2.152 | 1.877 | 2.895 | 423.2 | 22.0 | 417.9 | 392.2 | 484.6 | 658 |
| Annual | 2.193 | 0.224 | 2.134 | 1.848 | 3.331 | 419.4 | 23.9 | 414.0 | 375.9 | 536.9 | 8353 |





**Table 3.** Summary of $CH_4$ and $CO_2$ mixing ratios at Bode across four seasons during March 2013 to Feb 2014 [seasonal mean, one standard deviation (SD), median, minimum (Min.) and maximum (Max.)]

| Season | $CH_4$ (ppm) | | | | | $CO_2$ (ppm) | | | | |
|---|---|---|---|---|---|---|---|---|---|---|
| | Mean | SD | Median | Min. | Max. | Mean | SD | Median | Min. | Max. |
| Pre-Monsoon | 2.157 | 0.230 | 2.082 | 1.848 | 3.121 | 426.2 | 25.5 | 417.0 | 378.8 | 536.9 |
| Monsoon | 2.199 | 0.241 | 2.126 | 1.869 | 3.331 | 413.5 | 24.2 | 407.1 | 375.9 | 506.2 |
| Post-Monsoon | 2.210 | 0.200 | 2.167 | 1.879 | 2.762 | 417.3 | 23.1 | 414.1 | 381.9 | 486.7 |
| Winter | 2.214 | 0.209 | 2.177 | 1.877 | 2.895 | 421.9 | 20.3 | 419.3 | 386.7 | 494.5 |



**Table 4.** Comparison of monthly average $CH_4$ and $CO_2$ mixing ratios at a semi-urban and a rural site in Nepal (this study) with other urban and background sites in the region and elsewhere

| Site Setting | Bode, Nepal (Urban) | | | | Chanban, Nepal (Rural) | | Mauna Loa, USA (Background)[d] | | Waliguan, China (Background)[e] | |
|---|---|---|---|---|---|---|---|---|---|---|
| Species | $CO_2$ | $CH_4$ | $CO_2$ | $CH_4$ | $CO_2$ | $CH_4$ | $CO_2$ | $CH_4$ | $CO_2$ | $CH_4$ |
| Unit | ppm | ppb | Ppm | ppb | ppm | ppb | ppm | ppb | ppm | ppb |
| Mar 2013 | 426.6 | 2207.06 | | | | | 397.3 | 1839.82 | 399.5 | 1867.54 |
| Apr | 430.3 | 2183.30 | | | | | 398.4 | 1836.65 | 402.8 | 1874.03 |
| May | 421.7 | 2093.46 | | | | | 399.8 | 1833.66 | 402.5 | 1877.53 |
| Jun | 417.9 | 2060.91 | | | | | 398.6 | 1817.77 | 397.4 | 1887.36 |
| Jul | 410.3 | 2129.54 | | | | | 397.2 | 1808.36 | 393.3 | 1887.63 |
| Aug | 409.9 | 2274.34 | 411.3 | 2281.11 | 403.4 | 2049.71 | 395.2 | 1819.13 | 392.0 | 1892.78 |
| Sep | 414.9 | 2301.35 | 419.9 | 2370.93 | 399.1 | 2101.75 | 393.5 | 1835.79 | 393.1 | 1893.48 |
| Oct | 417.0 | 2210.02 | | | | | 393.7 | 1835.90 | 395.6 | 1876.36 |
| Nov | 417.2 | 2206.84 | | | | | 395.1 | 1834.49 | 397.1 | 1875.09 |
| Dec | 417.7 | 2205.91 | | | | | 396.8 | 1844.66 | 398.6 | 1880.21 |
| Jan 2014 | 424.8 | 2233.82 | | | | | 397.8 | 1842.20 | 398.8 | 1865.45 |
| Feb | 423.2 | 2199.01 | | | | | 397.9 | 1833.51 | 401.1 | 1877.64 |
| | | | | | | | | | | |
| *Annual* | | | | | | | | | | |
| Bode | 419.4 | 2193.07 | | | | | | | | |
| Mauna Loa | | | | | | | 396.8 | 1831.83 | | |
| Waliguan | | | | | | | | | 397.7 | 1879.59 |
| Nanjing (2011)[a] | 406.5 | | | | | | | | | |
| Shadnagar (2014)[b] | 394.0 | | | | | | | | | |
| Ahemadabad (2013-2015)[c] | 413.0 | 1920.0 | | | | | | | | |

[a] Huang et al., 2015, [b] Sreenivas et al., 2016, [c] Chandra et al., 2016, [d] ftp://aftp.cmdl.noaa.gov/data/trace_gases/ch4/in-situ/surface/mlo/; ftp://aftp.cmdl.noaa.gov/data/trace_gases/co2/in-situ/surface/mlo/, [e] ftp://aftp.cmdl.noaa.gov/data/trace_gases/co2/flask/surface/wlg/; ftp://aftp.cmdl.noaa.gov/data/trace_gases/ch4/flask/surface/wlg/



**Table 5**. Emission ratio of $CO/CO_2$ (ppb ppm$^{-1}$) derived from emission factors (gram of gas emitted from per kilogram of fuel burned, except transport sector which is derived from gram of gases emitted per kilometer distance travelled)

| Sectors | Details | $CO/CO_2$ | Reference |
|---|---|---|---|
| 1. Residential/Commercial | | | |
|   i. LPG | | 4.8 | Smith et al. (2000) |
|   ii. Kerosene | | 13.4 | Smith et al. (2000) |
|   iii. Biomass | | 52.9 - 98.5 | * |
|   iv. Diesel power | | | The World Bank |
|   generators | < 15 year old | 5.8 | (2014) |
| | >15 year old | 4.5 | |
| 2. Transport | | | ** |
| a. Diesel | | | |
|   i. HCV diesel bus | >6000cc, 1996-2000 | 4.9 | |
| | post 2000 and 2005 | 5.4 | |
|   ii. HCV diesel truck | >6000cc, post 2000 | 7.9 | |
| b. Petrol | | | |
|   i. 4 stroke motorcycle | <100 cc, 1996-2000 | 68 | |
| | 100-200 cc, Post 2000 | 59.6 | |
|   ii. Passenger cars | <1000 cc, 1996-2000 | 42.4 | |
|   iii. Passenger cars | <1000 cc, Post 2000 | 10.3 | |
| 3. Brick industries | | | |
|   i. BTK fixed kiln | | 17.2 | Weyant et al. (2014) |
|   ii. Clamp brick kiln | | 33.7 | Stockwell et al. (2016) |
|   iii. Zigzag brick kiln | | 3.9 | Stockwell et al. (2016) |
| 4. Open burning | | | |
|   i. Mixed garbage | | 46.9 | Stockwell et al. (2016) |
|   ii. Crop-residue | | 51.6 | Stockwell et al. (2016) |

\* Westerdahl et al. (2009)

\*\* http://www.cpcb.nic.in/Emission_Factors_Vehicles.pdf





**Table 6**. Seasonal average of the ratio of $dCO$ to $dCO_2$ over a period of 3 hours during (a) morning peak and (b) evening peak in the ambient mixing ratios of CO and $CO_2$

| Period | Season | $dCO/dCO_2$ (SD) | Median | N | Confidence interval (95%) |
|---|---|---|---|---|---|
| a. Morning hours (7:00-9:00) | Pre-monsoon | 7.6 (3.1) | 7.8 | 249 | 0.4 |
| | Monsoon | 2.2 (1.6) | 1.9 | 324 | 0.2 |
| | Post-monsoon | 3.1 (1.4) | 2.8 | 183 | 0.2 |
| | Winter* | 11.2 (4.4) | 11.0 | 255 | 0.5 |
| b. Evening hours (19:00-21:00) | Pre-monsoon | 15.1 (9.0) | 12.7 | 248 | 1.1 |
| | Monsoon | 8.0 (5.2) | 6.3 | 323 | 0.6 |
| | Post-monsoon | 11.5 (5.6) | 10.6 | 182 | 0.8 |
| | Winter | 21.6 (14.1) | 18.2 | 254 | 1.7 |
| c. Seasonal (all hours) | Pre-monsoon | 12.2 (13.3) | 8.8 | 1740 | 0.6 |
| | Monsoon | 7.5 (13.5) | 2.9 | 2176 | 0.6 |
| | Post-monsoon | 8.3 (12.4) | 4.4 | 1289 | 0.7 |
| | Winter | 15.1 (13.3) | 12.5 | 1932 | 0.6 |





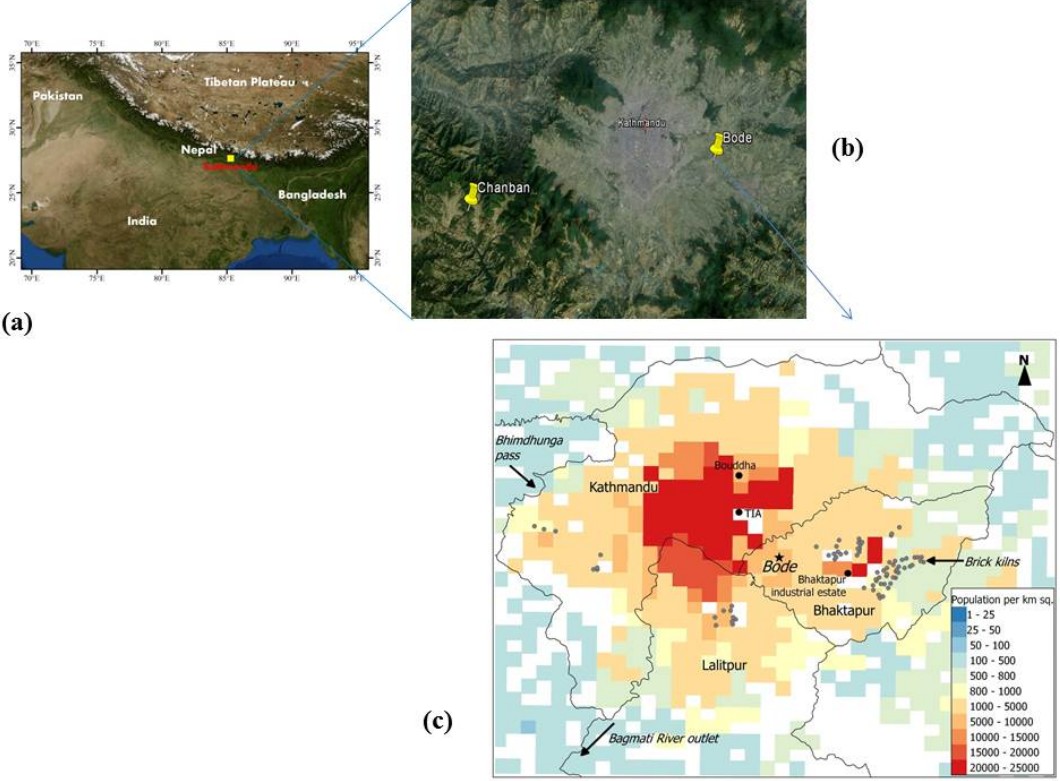

**Figure 1.** Location of measurement sites: (a) Kathmandu Valley (b) semi-urban measurement site at Bode in Kathmandu Valley, and a rural measurement site at Chanban in Makawanpur district Nepal, (c) general setting of Bode site. Colored grid and TIA represent population density and the Tribhuvan International Airport, respectively.




**Figure 2**. Time series of hourly average (a) mixing ratios of $CH_4$, $CO_2$, CO, and water vapor measured with a cavity ring down spectrometer (Picarro G2401) at Bode, and (b) temperature and rainfall monitored at the Tribhuvan International Airport (TIA), ~4 km to the west of Bode site in the Kathmandu Valley, Nepal. Temperature shown in pink color is observed at Bode site.









**Figure 3.** Monthly variations of the mixing ratios of hourly (a) $CH_4$, (b) $CO_2$, (c) CO, and (d) water vapor observed at a semi-urban site (Bode) in the Kathmandu Valley over a period of a year. The lower end and upper end of the whisker represents $10^{th}$ and $90^{th}$ percentile, respectively; the lower end and upper end of each box represents $25^{th}$ and $75^{th}$ percentile, respectively, and black horizontal line in the middle of each box is the median for each month while red dot represents mean for each month.











**Figure 4.** Pollution rose of the hourly $CH_4$ and $CO_2$ mixing ratios observed at Bode in the Kathmandu
Valley (a) $CH_4$ and (b) $CO_2$ from Mar 2013 to Feb 2014. Pollution rose shows variations of pollutants
based on frequency of counts by wind direction. The units of $CH_4$ and $CO_2$ are in ppm

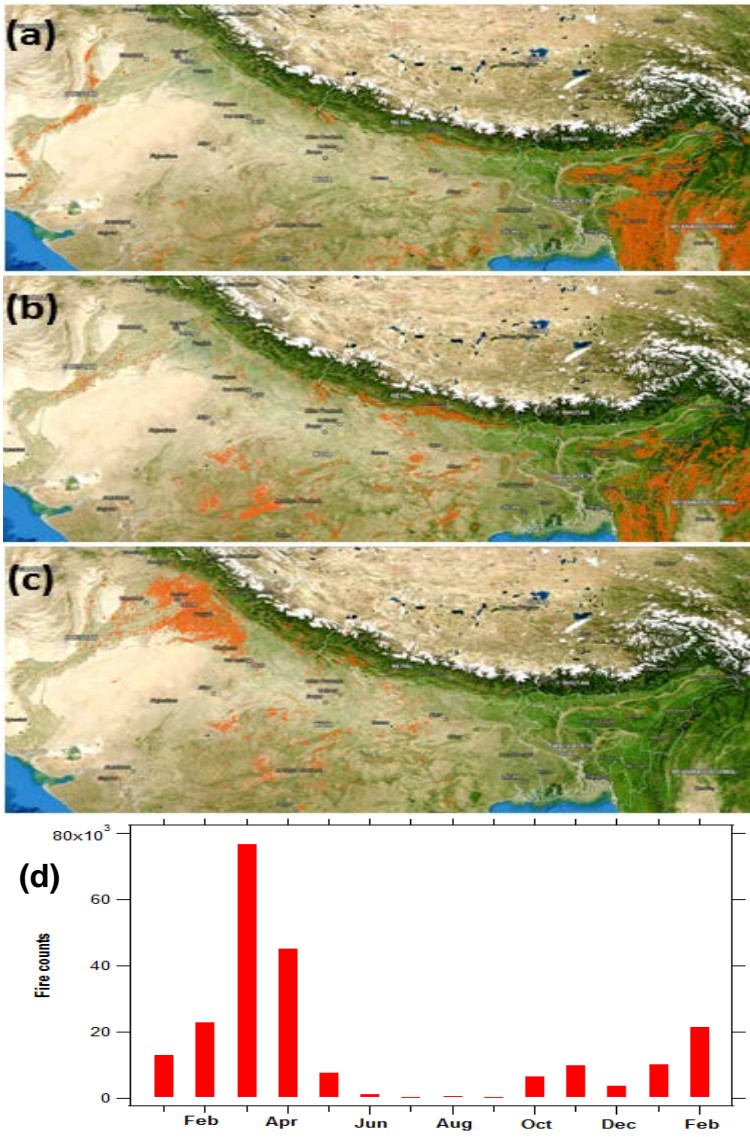

**Figure 5.** Satellite detected fire counts in (a) Mar, (b) Apr, (c) May 2013 in the broader region surrounding Nepal and (d) total number of fire counts detected by MODIS instrument onboard the Aqua satellite during Jan 2013-Feb 2014. Source: https://firms.modaps.eosdis.nasa.gov/firemap/




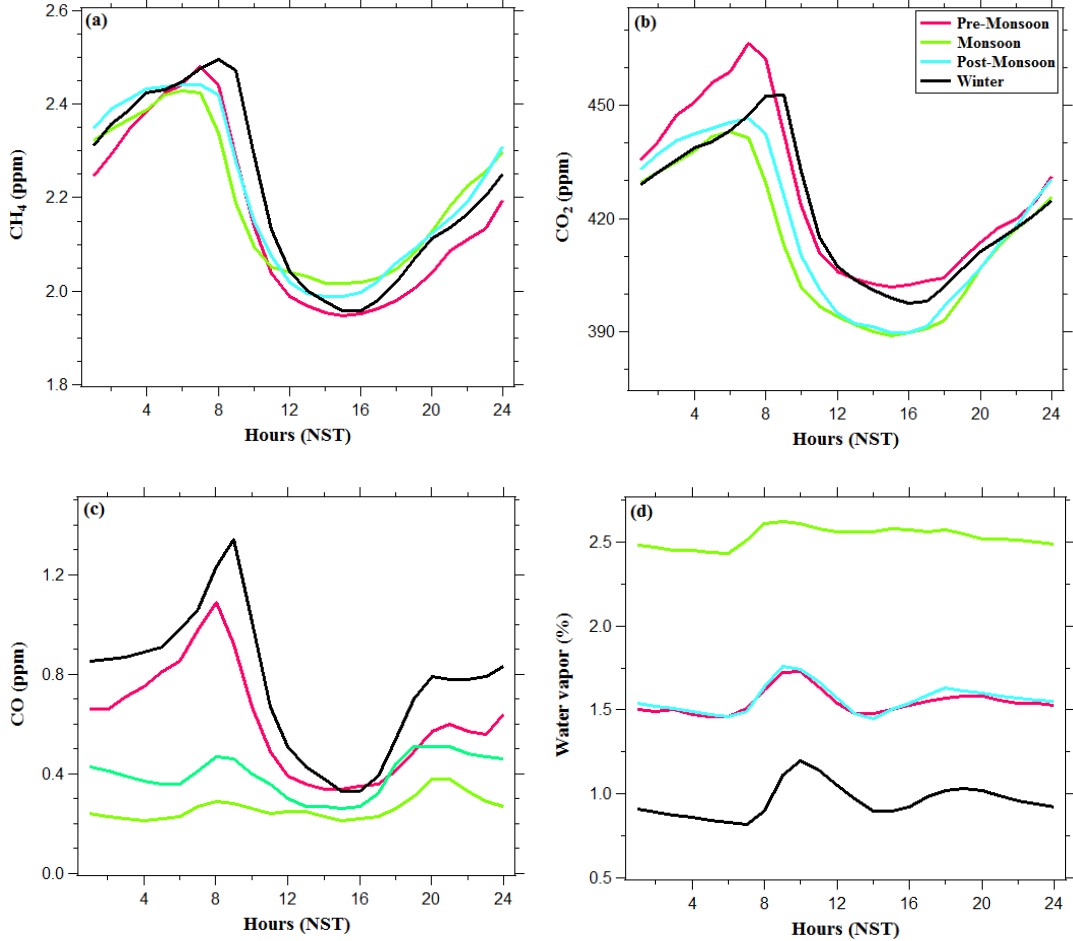

**Figure 6.** Diurnal variations of hourly mixing ratios in different seasons (a) $CH_4$, (b) $CO_2$, (c) CO, and (d) water vapor observed at Bode (semi-urban site) in the Kathmandu Valley during March 2013-February 2014. Seasons are defined as Pre-monsoon: Mar-May, Monsoon: Jun-Sep, Post-monsoon: Oct-Nov, Winter: Dec-Feb. The x axis is in Nepal Standard Time (NST).



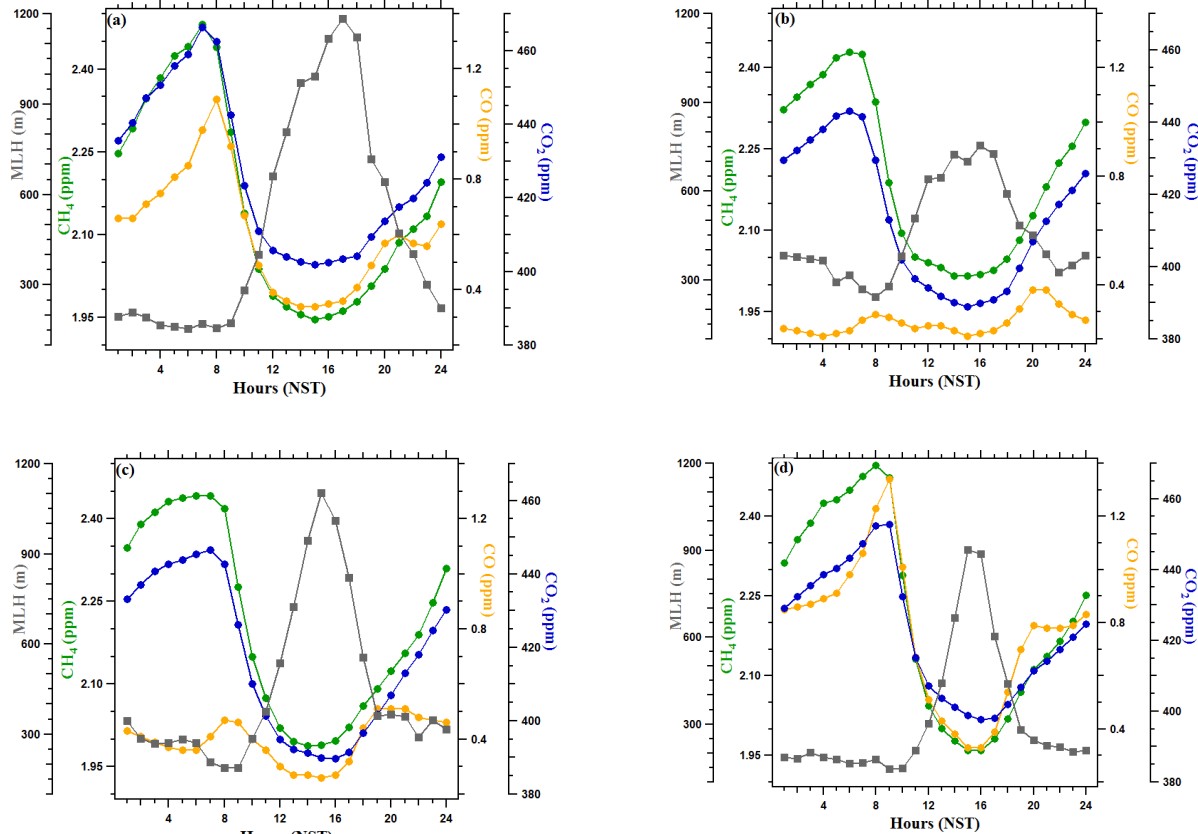





**Figure 7.** Diurnal variations of hourly mixing ratios of $CH_4$, $CO_2$, CO, and mixing layer height (MLH) at Bode (a semi-urban site in the Kathmandu Valley) in different seasons (a) pre-monsoon (Mar-May), (b) monsoon (Jun-Sep), (c) post-monsoon (Oct-Nov) and (d) winter (Dec-Feb) during March 2013- Feb 2014.





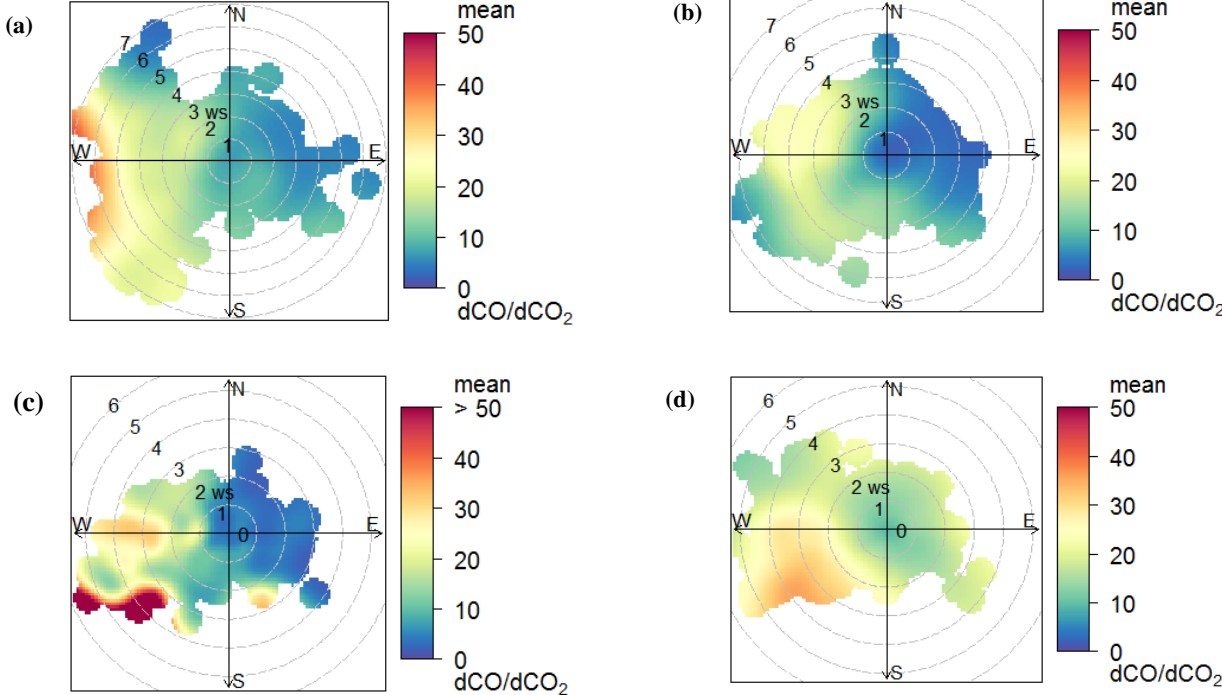

**Figure 8.** Seasonal polar plot of hourly $dCO/dCO_2$ ratio based upon wind direction and wind speed: (a) pre-monsoon, (b) monsoon, (c) post-monsoon and (d) winter seasons.





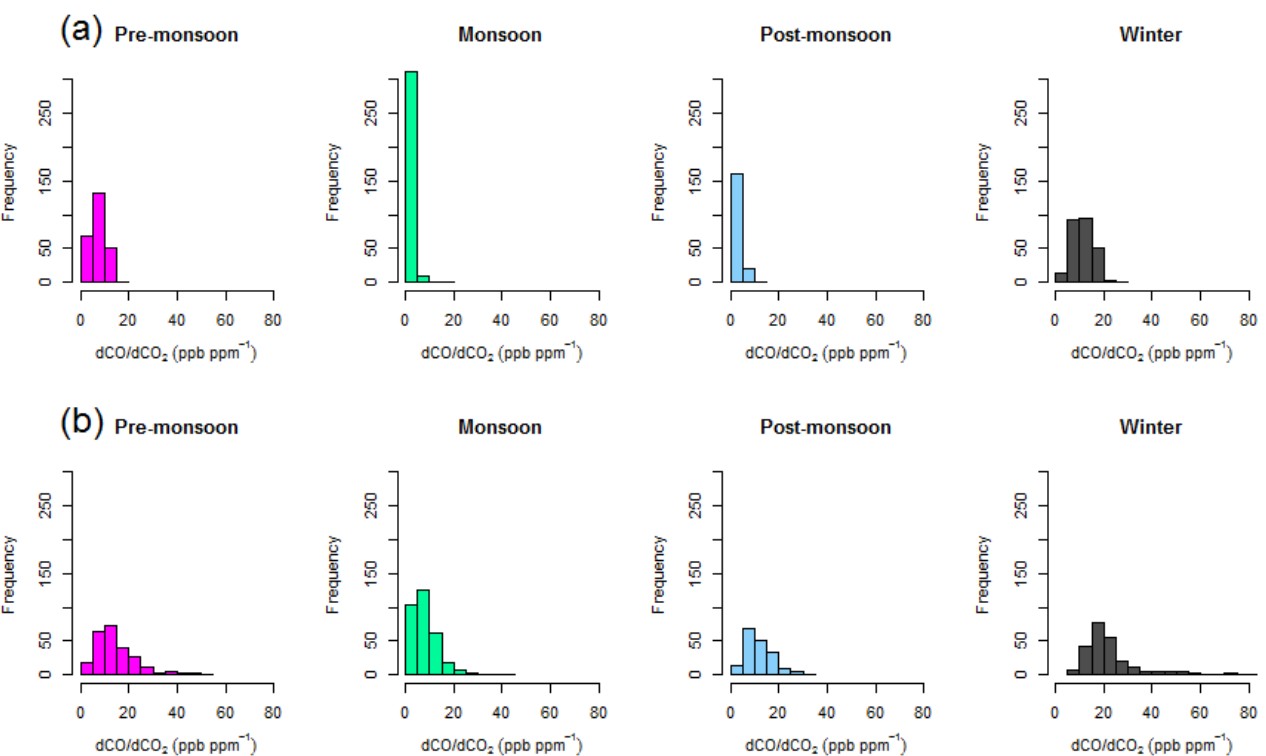

**Figure 9.** Seasonal frequency distribution of hourly $dCO/dCO_2$ ratio (a) morning hours (7:00-9:00) in all season except winter (8:00-10:00), (b) evening hours (19:00-21:00)





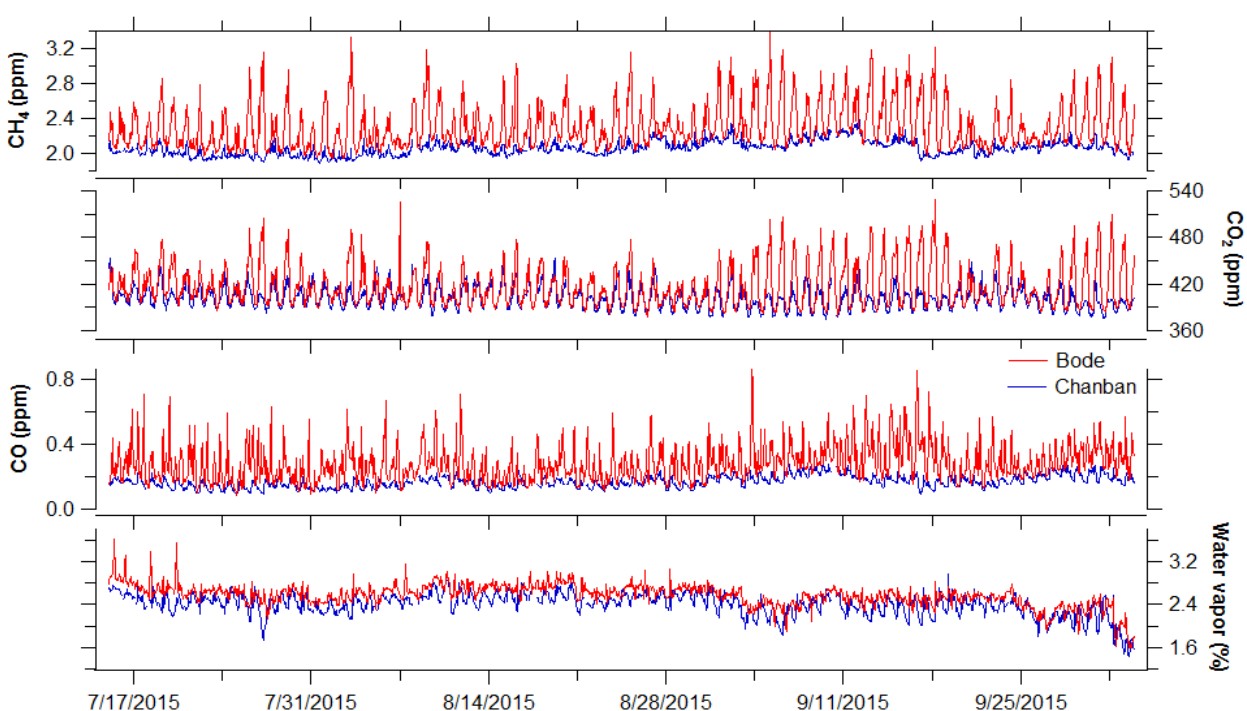

**Figure 10.** Comparison of hourly average mixing rations of $CH_4$, $CO_2$, CO, and water vapor observed at Bode (a semi-urban site) in the Kathmandu Valley and at Chanban (a rural/background site) in Makawanpur district, ~ 20 km from Kathmandu, on other side of a tall ridge.



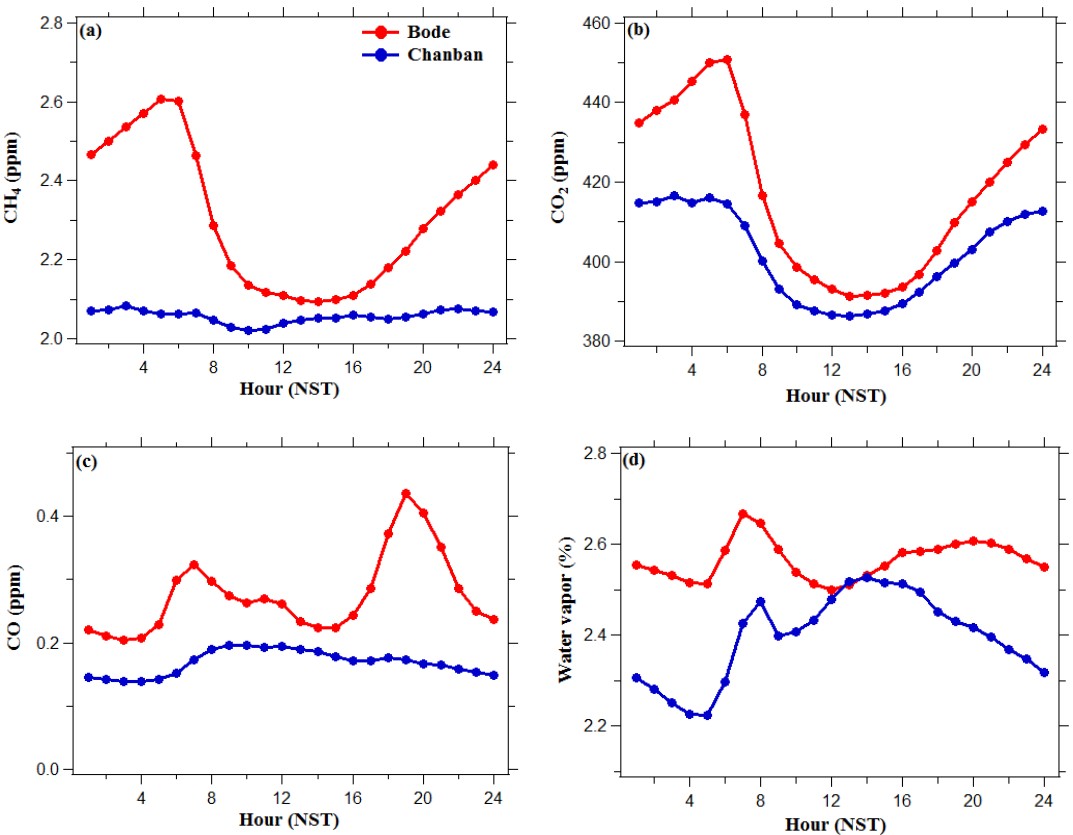

**Figure 11.** Diurnal variations of hourly average mixing ratios of (a) $CH_4$, (b) $CO_2$, (c) CO and (d) water vapor observed at Bode in the Kathmandu Valley and at Chanban in Makawanpur district during 15 July- 03 October 2015