# Peer review of "Seasonal and diurnal variations of methane and carbon dioxide in the Kathmandu Valley"

_Atmospheric Chemistry and Physics, 2016_

## Referee Comment (RC1) · Anonymous Referee #1 · 13 Apr 2017

General comments: Atmospheric greenhouse gases (GHGs) such as CO2, CH4, H2O and CO are important climate forcing agents having significant impacts on climate system and air quality. This study brings outs first continuous measurements of atmospheric GHGs using high precision cavity ring down spectrometer (Picarro 24G2401, USA) at Kathmandu Valley during March 2013 to March 2014. The authors have done and extensive study on GHGs variability with time and space. However, there are a few minor technical changes in the manuscript. This paper is recommend to publish in ACP after incorporating the minor technical corrections. Line18-20: This paper studies about GHGs and GHGs are not classified as pollutants especially CO2 and CH4. Impact of pollution on GHGs need to be emphasis not to refer GHGs as

pollutants. Also sentence "This paper reports. . .. May be re-written Line51-53: Not clear. Authos may please check the sentence. " All three species showed strong diurnal and saying immediately CH4 and CO did not show any variation. . .May be provided quantitative numbers. Line139: Rupakheti et al., 2016 need to be updated if available Line252: Sentence "The % may be written as Difference (%) of the analyzer differed by. . .. Line349: Units of GHGs and other gases should be uniform in the manuscript. Line395-396: Impact of rainfall on CO2 dilution process may be supported with reference Mahesh et al., 2014 "Impact of land-sea breeze. . .. Line432-433: Please check the statement that CO2 will be high but CH4 will be high during post-monsoon season. Figure4: Titled should be changed. Since GHGs are not pollutants and legend should be CH4 not Ch4 Figure11: Show double Y-axis for better visualisation

Please also note the supplement to this comment:
http://www.atmos-chem-phys-discuss.net/acp-2016-1136/acp-2016-1136-RC1-supplement.pdf
* * *

---

## Referee Comment (RC2) · Anonymous Referee #2 · 7 May 2017

Highly precise and long-term measurements of greenhouse gases are essential to understand the underlying processes in the context of global climate change. It is particularly valuable in regions like Asia where it is currently limited by dedicated long term observations. This study demonstrates the observational variations of $CO_2$ and $CH_4$ mixing ratios in the Kathmandu Valley (Nepal), and compares them to those of other rural sites. The scope of this study is hence highly relevant to the public and authors' efforts on this regard need to be appreciated. However, the manuscript in its present form does not meet the standard to merit the publication in ACP, and needs to be adequately revised. I therefore recommend the current manuscript to undergo major revision to be considered in ACP.

[Figure]

General comments

As mentioned above, I value authors' effort in this study as an important step towards generating observational wealth which can be tremendously used by scientific community to understand a wide range of mechanisms involved in this aspect. Given the complicated interplay of many processes involved, single approach cannot answer the unresolved scientific questions related to these processes and mechanisms affecting mixing ratio variations. This requires defining far more systematic approaches and other robust tools. However the finding from this study can be very valuable if presented with adequate measurement and analysis methods/techniques used, including a good summary of the methods and a much clear report on the observational variations of these analysed tracers. The results will be more convincing by focusing on the key aspects of the data rather than trying to relate them to flux categories based on assumption (many times) and without sufficient tools (inverse modeling). In some places, results are presented nicely, but then an explanation is suggested based on other literature that focussed on other study regions/methods, without even demonstrating its relevance to this dataset. Also in some places, the study gets into overly ambitious interpretations of the results and conclusions on the basis of single analysis. On this basis, I recommend authors to focus on presenting this study by clearly stating the measurement and analysis techniques used, defining their strategies, providing analysed results & possible uncertainties and giving more convincing interpretations of the results and conclusions. I highly recommend authors not to jump to interpreting emission/flux sources and patterns based on the single site measurements and the analysis done in this study. Sections in this manuscript dealing with these aspects needs to be (preferably) removed or highly restructured.

Specific comments and suggestions for revised analysis

Page 3, lines 63-66 "Between 1750 . . . .cover changes" Misleading sentence. The given estimation is overly high for the accumulated $CO_2$ in the atmosphere. As per IPCC reports & other studies, it is in the range of 230-250 Pg C. The given number is more

towards total (cumulative) CO2 emissions between 1750 and 2011, which were partly compensated by the ocean and terrestrial ecosystems.

Page 4, line 86-89 "important sources" Give reference

Page 4, lines 90-92 "Ecosystem and ..... between July-October Please give appropriate reference for this statement. Prasad et al., 2014 investigated based on satellite GHG concentration observations, not based on inverse or ecosystem models. In my knowledge there's no such inverse flux estimations available over South Asia, accounting (and decoupling) seasonal variations of CO2 uptake and release to the atmosphere. However, Patra et al., 2011 showed inverse estimations of monthly co2 fluxes over South Asia. Please correct it.

Page 12, lines 308-310 "2.193 ($\pm$ 0.224) ppm, 419.4 ($\pm$ 23.9) ppm, 0.50 ($\pm$ 0.35) ppm, and 310 1.71 ($\pm$ 0.71) %" Uncertainty seems to be quite high for co2, ch4 and co. Why? Please include the reason in the text.

Page 12, lines 318-320 "CH4 was ..... observation period" Given the above uncertainty ranges of Bode values (nearly 10.2% for CH4, 5.7% for CO2 and 70% for CO),the estimated percentages of increment relative to other observatories are also biased. These uncertainties need to be take into account, or at least properly mentioned.

Page 12, lines 320-322 "The small ..... Asia region"

Although I tend to agree that we can expect (+ seeing satellite images), most of the cases, higher CH4 mixing ratios in Asia relative to Mauna Loa observatory, authors should note that this conclusion, as given in the text, about the whole Asia cannot be drawn from analysing "just two Asian sites" in a given time period. This sentence is misleading, and needs to be reformulated.

Page 12, line 324 "Ahmedabad (1.880 ppm) (Sahu and Lal, 2006) and Shadnagar (1.92 $\pm$ 0.07)"

See my above comment on uncertainty.

Page 12, lines 326-332 "Likewise, the …...  in China" These estimated increments are meaningless based on the uncertainty range of Bode's tracer mixing ratios. That is, these 5.7% and 5.5 % increments are statistically insignificant when it is compared with CO2 values with 5.7% uncertainty range. I strongly recommend authors to remove this.

Page 13, lines 354-355 "burning activities due to rainfall in the region"

What about the burning activities in Bode area during this rainy period of time?

Page 14, lines 384-385 "The seasonal …...  residue burn" What about seasonality of atmospheric transport and other met. fields? I would think that it can also make a significant impact especially considering mountain valley effect.

Page 14, lines 388 "partially due to rain washout. " It can also very well due to relatively high advection and vertical mixing.

Page 15, lines 395-396 "related to less or no rainfall, which results in the absence of rain washout" if it is with transport or raining, it should also affect CH4 and other tracers. Please clarify.

Page 15, lines 412-413 "which had > 2.5 ppm CH4 and > 450 CO2" It's likely that the increase in CO2 & CH4 is associated with advected signals from the North East and the East; however it is not clear how it can be interpreted as the advected airmass had the said values for CO2 and CH4 unless the study used any tracer transport and emission models. Wind direction and tracer concentration from the given site alone are not sufficient to conclude this. A clarification is needed here. Otherwise I recommend authors to remove this.

Page 15, lines 414 "(not shown in Figure 4)" I encourage authors to show CO as well for the completeness of the interpretation.

Page 15, lines 417 "high CH4 emissions" What about CO2?

 Page 17, line 455 "The westerly circulation (originated at longitude about 60E in 5 days back trajectories)" What does it mean? What's originated? Talking about model? Based on modeled trajectories?

Page 17, lines 475-476 "$CO_2$ mixing ratios whereas CO shows an  evening peak" It's surprising. Why is it so? Please clarify.

Page 19, lines 519-521 "While the .. ... other seasons" I couldn't follow how it's related. A clarification is highly needed. What are these other most CO sources mentioned here?

Page 20, lines 551-559 "Highest day .. ...Mauna Loa and Waliguan" It's lost. I see many assumptions here rather than convincing statements. What about biospheric activity and its seasonality? mesoscale transport mechanisms?

Page 21, lines 577-579 "Overall, the .. ... fire etc." Importantly it shares the transport mechanisms.

Pages 21-22, Sec. 3.5 I strongly recommend authors to remove the whole section. It is not at all straight forward, as assumed here, to determine the impact of emission sources and transport, based on concentration measurements from single site. It does not make any sense unless a dedicated further study is involved to justify the stated assumptions here. The section, as in the present shape, does not meet scientific reasoning; hence need to be removed.

Page 23, lines 633-641 "Based on the .. ... post-monsoon seasons" This could be a likely scenario. Have these interpretations been supported by any emission inventories available? It is also important to point out the associated uncertainties involved in separating different emission sectors based on this approach. Note that this approach cannot separate near and far field sources, different lifetimes of tracers etc.

Page 25, lines 704-706 "but it is clear that" What makes it clear?

Page 26, lines 731-732 "Regional transport .. ... during pre-monsoon"   I don't see

any valid justification for this throughout.

Page 27, lines 743-744 "Low values ..... mixing ratios." Please provide supporting details.

Page 27, line 746 "useful for evaluation of satellite measurements
climate" How?

Page 27, lines 747-749 "The analysis ..... Kathmandu
Valley.
" Please remove this sentence. Note that this is not met here and the study only demonstrates the observational variations of GHGs in the study region.

Reference Please check. Formatting issues and sometimes journal details (or other important parts) are missing

Table 4 Columns 4 & 5 : What are these values? Looks like monthly values for Bode during Aug., and Sep. 2013; but then why are they different from corresponding 2nd column of the table 4?

Table 6 Column 2: This "*" meant for?

Figure 4 I didn't follow the fig. well. What I understood is that the plot shows the frequency of hourly mixing ratios w.r.t the frequency of prevailing wind direction. Did it also take into account the wind speed? Then what about different percentages shown? For example, does it mean that 5% of sample time in August, the wind was from NE and "CH4_corrected" is above 2.5 ppm in which less than 1% time (in my eyes), CH4 is in 3-3.3 ppm range? In that case, it's statistically difficult to say that the monthly enhancement is due to the polluted air masses from the NE and E. By the way, what are these "_corrected" values for CH4 and CO2? What about March-April scenario for CO2_corrected? Did two plots (Figs.2 and 3) use same set of master data, or any quality filtering had been done other than monthly averaging?

---

## Author Comment (AC1) · 2 Jul 2017

July 02, 2017

Dear Editor,

We would like to thank you for serving as the editor for our manuscript. We have revised the manuscript to incorporate all of the general and specific comments and suggestions from the two reviewers. As the reviewers have suggested, we have removed section 3.5 and some sentences in other sections that appeared speculative, where we only had weak evidence to justify our arguments. Similarly, we rephrased several sentences and made them more understandable by adding further information. We believe that with these revisions the interpretation of the findings and hence the quality of the manuscript has improved significantly.

Sincerely yours,

Khadak Singh Mahata on behalf of all coauthors

**Seasonal and diurnal variations of methane and carbon dioxide in the Kathmandu Valley in the foothills of the central Himalaya**

by Khadak Singh Mahata et al., 2017 (ACPD)

We would like to thank both reviewers for their constructive comments and suggestions. Please find below the reviewers' comments in black, our response in blue and changes in the revised manuscript in red. The line numbers in our response refer to the line numbers in the revised manuscript.

**Reviewer 1**

General comments: Atmospheric greenhouse gases (GHGs) such as CO2, CH4, H2O and CO are important climate forcing agents having significant impacts on climate system and air quality. This study brings outs first continuous measurements of atmospheric GHGs using high precision cavity ring down spectrometer (Picarro 24G2401, USA) at Kathmandu Valley during March 2013 to March 2014. The authors have done and extensive study on GHGs variability with time and space. However, there are a few minor technical changes in the manuscript. This paper is recommend to publish in ACP after incorporating the minor technical corrections.

We would like to thank reviewer for considering that the manuscript contains extensive work on the variability of GHGs in the Kathmandu Valley. We have incorporated the reviewer's comments and suggestions to the extent possible in the following sections of the manuscript.

Line18-20: This paper studies about GHGs and GHGs are not classified as pollutants especially CO2 and CH4. Impact of pollution on GHGs need to be emphasis not to refer GHGs as pollutants. Also sentence "This paper reports …". May be re-written

We agree with the reviewer that $CO_2$ and $CH_4$ at normal ambient concentrations are not considered as pollutants from the health perspective. Thus, we did not describe them as pollutants in the manuscript. We slightly modified the sentence "this paper reports…" to distinguish the GHGs from the pollutant CO, and thus more understandably reflect what has been looked into in this study.

Line51-53: Not clear. Authors may please check the sentence. " All three species showed strong diurnal and saying immediately CH4 and CO did not show any variation: : :May be provided quantitative numbers.

Thank you for pointing out that the statement was not clear to the reviewer. Here we are explaining the diurnal variations of $CH_4$, $CO_2$ and CO at two measurement sites (Bode and Chanban). We have articulated it by rephrasing the sentence as follows (lines 52-56):

At Bode, all three gas species ($CO_2$, $CH_4$ and CO) showed strong diurnal patterns in their mixing ratios with a pronounced morning peak (ca. 08:00), a dip in the afternoon, and again gradual increase through the night until the next morning, whereas $CH_4$ and CO mixing ratios at Chanban did not show any noticeable diurnal variations.

Line139: Rupakheti et al., 2016 need to be updated if available

The Rupakheti et al. manuscript will be submitted to ACPD soon. We have updated the citation as Rupakheti et al. (2017, manuscript in preparation) in line 144-145.

Line252: Sentence "The % may be written as Difference (%) of the analyzer differed by…".

Corrected. The sentence has been rephrased as follows (lines 258-260):

The difference between $CO_2$ mixing ratio reported by the analyzer and the reference mixing ratio was within 5%.

Line349: Units of GHGs and other gases should be uniform in the manuscript.

Corrected. They are now reported in ppm throughout the manuscript.

Line395-396: Impact of rainfall on CO2 dilution process may be supported with reference Mahesh et al., 2014 "Impact of land-sea breeze…".

Frequent rainfall suppresses emission sources which results in reducing mixing ratios of the gas and aerosol species. To reflect this, we have modified the sentences in lines 414-422:

The concentrations of most pollutants in the region are lower during the monsoon period (Sharma et al., 2012, Marinoni, 2013; Putero et al., 2015) because frequent and heavy rainfall suppresses emission sources. We saw a drop in the $CO_2$ mixing ratio during the rainfall period due to changes in various processes such as enhanced vertical mixing, uptake of $CO_2$ by vegetation and soils, and where relevant reduction in combustion sources. $CO_2$ can also dissolve into rainfall, forming carbonic acid, which may lead to a small decrease in the $CO_2$ mixing ratio as has been observed during heavy intensity rainfall (Mahesh et al., 2014; Chaudhari et al., 2007).

The following references are added in the list of references.

Mahesh P, Sharma N, Dadhwal VK, Rao PVN, Apparao BV, et al.:Impact of Land-Sea Breeze and Rainfall on CO2Variations at a Coastal Station. J. Earth Sci. Clim. Change, 5:201. doi: 10.4172/2157-7617.1000201, 2014.

Chaudhari, P.R., Gajghate, D.G., Dhadse, S. et al. Environ. Monit. Assess., 135: 281. doi:10.1007/s10661-007-9649-7, 2007.

Line432-433: Please check the statement that CO2 will be high but CH4 will be high during post-monsoon season.

This comment does not match with the lines mentioned by the reviewer. We did not find the exact or similar sentence in the manuscript.

Figure 4: Titled should be changed. Since GHGs are not pollutants and legend should be CH4 not Ch4.

In order to remove confusion, we have changed the caption of Figure 4 as follows:

Relation between mixing ratios and wind direction observed at Bode in the Kathmandu Valley (a) $CH_4$,(b) $CO_2$, and (c) CO from March 2013 to February 2014. The figure shows variations of $CH_4$, $CO_2$ and CO mixing ratios based on frequency counts of wind direction (in %) as represented by circle. The color represents the different mixing ratios of the gaseous species. The units of $CH_4$, $CO_2$ and CO are in ppm.

The legend is also corrected as suggested.

Figure11: Show double Y-axis for better visualization

Thank you for the suggestion. As individual species varies over a wide range and there is also an order of magnitude difference in their mixing ratios, putting them in double Y-axes with different scales makes the figure more confusing. We have therefore kept the figure as it is.

**Reviewer 2**

Highly precise and long-term measurements of greenhouse gases are essential to understand the underlying processes in the context of global climate change. It is particularly valuable in regions like Asia where it is currently limited by dedicated long term observations. This study demonstrates the observational variations of CO2 and CH4 mixing ratios in the Kathmandu Valley (Nepal), and compares them to those of other rural sites. The scope of this study is hence highly relevant to the public and authors' efforts on this regard need to be appreciated. However, the manuscript in its present form does not meet the standard to merit the publication in ACP, and needs to be adequately revised. I therefore recommend the current manuscript to undergo major revision to be considered in ACP.

We would like to thank reviewer for considering our measurements as important for the region and that the scope of our study is relevant to the public. We have revised the manuscript as suggested by the reviewer.

General comments

As mentioned above, I value authors' effort in this study as an important step towards generating observational wealth which can be tremendously used by scientific community to understand a wide range of mechanisms involved in this aspect. Given the complicated interplay of many processes involved, single approach cannot answer the unresolved scientific questions related to these processes and mechanisms affecting mixing ratio variations. This requires defining far more systematic approaches and other robust tools. However the finding from this study can be very valuable if presented with adequate measurement and analysis methods/techniques used, including a good summary of the methods and a much clear report on the observational variations of these analysed tracers. The results will be more convincing by focusing on the key aspects of the data rather than trying to relate them to flux categories based on assumption (many times) and without sufficient tools (inverse modeling). In some places, results are presented nicely, but then an explanation is suggested based on other literature that focussed on other study regions/methods, without even demonstrating its relevance to this dataset. Also in some places, the study gets into overly ambitious interpretations of the results and conclusions on the basis of single analysis. On this basis, I recommend authors to focus on presenting this study by clearly stating the measurement and analysis techniques used, defining their strategies, providing analysed results & possible uncertainties and giving more convincing interpretations of the results and conclusions. I highly recommend authors not to jump to interpreting emission/flux sources and patterns based on the single site measurements and the analysis done in this study. Sections in this manuscript dealing with these aspects needs to be (preferably) removed or highly restructured.

Thank you for the general comments. As per the specific comments below, the analysis techniques are described better. We have focused on the key aspect of the data: (i) better interpretation of the results, as suggested by the reviewer, and (ii) removed speculative sentences and the whole section 3.5, which reviewer found less convincing. Based on our understanding of the referee's comments, we hope that by addressing all the specific comments collectively we will have adequately addressed the referee's general comments as well.

Specific comments and suggestions for revised analysis

Page 3, lines 63-66 "Between 1750 : : :.cover changes" Misleading sentence. The given estimation is overly high for the accumulated CO2 in the atmosphere. As per IPCC reports & other studies, it is in the range of 230-250 Pg C. The given number is more towards total (cumulative) CO2 emissions between 1750 and 2011, which were partly compensated by the ocean and terrestrial ecosystems.

We would like to thank reviewer for noticing the misleading sentence which is now corrected with $CO_2$ accumulation data from the IPCC 2013 report as follows (lines 66-67):

Between 1750 and 2011, 240(±10) PgC of anthropogenic $CO_2$ was accumulated in the atmosphere…. (IPCC, 2013).

Page 4, line 86-89 "important sources" Give reference

The following papers have been cited (see changed line 90-94) and the full citation is also included in the reference section.

Central Bureau of Statistics (CBS).: Nepal Living Standards Survey 2010/11, Statistical Report Volume 1, Central Bureau of Statistics, Government of Nepal, 2011.

Pandey, A., Sadavarte, P., Rao A. B., Venkataraman, C.: Trends in multipollutant emissions from a technology-linked inventory for India: II. Residential, agricultural and informal industry sectors, Atmos. Environ., 99, 341-352, doi: 10.1016/j.atmosenv.2014.09.080, 2014.

Sinha, V., Kumar, V., and Sarkar, C.: Chemical composition of pre-monsoon air in the Indo-Gangetic Plain measured using a new air quality facility and PTR-MS: high surface ozone and strong influence of biomass burning, Atmos. Chem. Phys., 14, 5921-5941, 10.5194/acp-14-5921-2014, 2014.

Page 4, lines 90-92 "Ecosystem and : : :.. between July-October Please give appropriate reference for this statement. Prasad et al., 2014 investigated based on satellite GHG concentration observations, not based on inverse or ecosystem models. In my knowledge there's no such inverse flux estimations available over South Asia, accounting (and decoupling) seasonal variations of CO2 uptake and release to the atmosphere. However, Patra et al., 2011 showed inverse estimations of monthly co2 fluxes over South Asia. Please correct it.

Thank you for pointing out the incorrect statement. The sentences and reference have been replaced by relevant reference in lines 95-100 as follows:

By using inverse modeling, Patra et al. (2011) found a net $CO_2$ uptake ($0.37 \pm 0.20$ Pg C $yr^{-1}$) during 2008 in South Asia and the uptake (sink) is highest during July-September. The remaining months act as a weak gross sink but a moderate gross source for $CO_2$ in the region.

Page 12, lines 308-310 "2.193 (_ 0.224) ppm, 419.4 (_ 23.9) ppm, 0.50 (_ 0.35) ppm, and 1.71 (_ 0.71) %" Uncertainty seems to be quite high for co2, ch4 and co. Why? Please include the reason in the text.

We apologize for not clarifying about the values in the parenthesis adjacent to the annual average value for the three species. They are not measurement uncertainties. Instead, they are one standard deviation, calculated from the hourly data for the observation period (of one year). The reported annual mean from the references sites such as Mauna Loa and referred to in this paper were calculated from monthly means, not hourly data. Therefore for consistency and ease of comparison, we have now reported annual average and the standard deviation from monthly mean data for all sites. Please see the correction (lines 316-322):

For the entire sampling period, the annual average ($\pm$ one standard deviation) of $CH_4$, $CO_2$, CO, and water vapor mixing ratios at Bode were 2.192 ($\pm$ 0.066) ppm, 419.3 ($\pm$ 6.0) ppm, 0.50 ($\pm$ 0.23) ppm, and 1.73 ($\pm$ 0.66)% respectively. The relative variabilities for $CH_4$, $CO_2$ and CO were thus 3%, 1.4% and 46%, respectively. Their variabilities at Mauna Loa were CH4: 6% and $CO_2$: 0.5% and at Waligaun were $CH_4$: 0.48%, $CO_2$: 0.9%. The high variability in the annual mean, notably for CO in Bode could be indicative of the seasonality of emission sources and meteorology.

Page 12, lines 318-320 "CH4 was : : :.. observation period" Given the above uncertainty ranges of Bode values (nearly 10.2% for CH4, 5.7% for CO2 and 70% for CO),the estimated percentages of increment relative to other observatories are also biased. These uncertainties need to be taken into account, or at least properly mentioned.

As noted above, these are not uncertainties, rather variabilities. We have recalculated the annual average and standard deviation values based on monthly average data. Now, the variabilities for $CH_4$, $CO_2$ and CO at Bode are 3%, 1.4% and 46%, respectively. The comparison between Bode and the other sites (Mauna Loa and Waligaun) for the GHGs now looks more reasonable. The following text has been inserted to reflect their variabilities and statistical significance (lines 330-334)

We performed a significance test at 95% confidence level (t-test) of the annual mean values between the sites to evaluate whether the observed difference is statistically significant ($p <$ 0.05), which was confirmed for the annual mean $CH_4$ and $CO_2$ between Bode and Mauna Loa, and between Bode and Waliguan.

Page 12, lines 320-322 "The small : : :.. Asia region" Although I tend to agree that we can expect (+ seeing satellite images), most of the cases, higher CH4 mixing ratios in Asia relative to Mauna Loa observatory, authors should note that this conclusion, as given in the text, about the whole Asia cannot be drawn from analysing "just two Asian sites" in a given time period. This sentence is misleading, and needs to be reformulated.

Thanks for drawing our attention to it. Yes, we didn't intend to generalize our comparison for Asia. We rephrased the following paragraph (lines 334-340):

$CH_4$ was nearly 20% higher at Bode than at Mauna Loa (1.831 ± 0.110 ppm) (Dlugokencky et al., 2017) and ca.17% higher than at Mt. Waliguan (1.879 ± 0.009 ppm) for the same observation period (Dlugokencky et al., 2016). The slightly higher $CH_4$ mixing ratios at Bode and Waliguan than at Mauna Loa Observatory could be due to prevalence of rice farming as a key source of $CH_4$ in this part of Asia.

Page 12, line 324 "Ahmedabad (1.880 ppm) (Sahu and Lal, 2006) and Shadnagar (1.92 _ 0.07)" See my above comment on uncertainty.

We have calculated the variabilities for Bode, Ahmedabad and Shadnagar, and rephrased the sentences in lines 340-345 as follows:

Similarly, the annual average $CH_4$ at Bode during 2013-14 was found comparable to an urban site in Ahmedabad (1.880 ± 0.4 ppm, i.e., variability: 21.3%) in India for 2002 (Sahu and Lal, 2006) and 14% higher than in Shadnagar (1.92 ± 0.07 ppm, i.e., variability: 3.6%), a semi-urban site in Telangana state (~70 km north from Hyderabad city) during 2014 (Sreenivas et al., 2016).

Page 12, lines 326-332 "Likewise, the : : :.. in China" These estimated increments are meaningless based on the uncertainty range of Bode's tracer mixing ratios. That is, these 5.7% and 5.5 % increments are statistically insignificant when it is compared with CO2 values with 5.7% uncertainty range. I strongly recommend authors to remove this.

See above regarding the confusion between uncertainty and variability, for which we apologize that this was not clear. We have recalculated the standard deviations based on the monthly values, and have also conducted a t-test between the annual mean mixing ratios and found that differences between the means at Bode and Mauna Loa, and between Bode and Waliguan were statistically significant. We have removed unnecessary explanations and inserted the following sentence (lines 345-348):

Likewise, the difference between annual mean $CO_2$ mixing ratios at Bode (419.2 ±6.0 ppm, 1.4% variability) vs. Mauna Loa (396.8 ± 2.0 ppm, 0.5% variability) (NOAA, 2015) and Bode vs. Waliguan (397.7 ± 3.6 ppm, 0.9% variability) (Dlugokencky et al., 2016a) is statistically significant ($p < 0.05$).

Page 13, lines 354-355 "burning activities due to rainfall in the region" What about the burning activities in Bode area during this rainy period of time?

Similar to Chanban, the burning activities around Bode area were also reduced or absent during the rainy season. We added the following line (378-379) in the manuscript.

The garbage and agro-residue burning activities were also absent or reduced around Bode due to rainfall during the monsoon period.

Page 14, lines 384-385 "The seasonal : : :... residue burn" What about seasonality of atmospheric transport and other met. fields? I would think that it can also make a significant impact especially considering mountain valley effect.

Agree. The paragraph has been restructured in lines 408-414 as:

The seasonal variation in $CO_2$ could be due to (i) the seasonality of major emission sources such as brick kilns, (ii) seasonal growth of vegetation ($CO_2$ sink) (Patra et al., 2011) and (iii) atmospheric transport associated with regional synoptic atmospheric circulation (monsoon circulation and westerly disturbance in spring season) which could transport regional emission sources from vegetation fires and agriculture residue burning (Putero et al., 2015), and a local mountain-valley circulation effect (Kitada and Regmi, 2003; Panday et al., 2009).

Page 14, lines 388 "partially due to rain washout. " It can also very well due to relatively high advection and vertical mixing.

Thank you for pointing out other possible reasons of low mixing ratios of gas species in rainy season. They are included (lines 414-422):

The concentrations of most pollutants in the region are lower during the monsoon period (Sharma et al., 2012, Marinoni, 2013; Putero et al., 2015) because frequent and heavy rainfall suppresses emission sources. We saw a drop in the $CO_2$ mixing ratio during the rainfall period due to changes in various processes such as enhanced vertical mixing, uptake of CO2 by vegetation and soils, and where relevant reduction in combustion sources. $CO_2$ can also dissolve into rainfall, forming carbonic acid, which may lead to a small decrease in the $CO_2$ mixing ratio as has been observed during heavy intensity rainfall (Mahesh et al., 2014; Chaudhari et al., 2007).

Page 15, lines 395-396 "related to less or no rainfall, which results in the absence of rain washout" if it is with transport or raining, it should also affect CH4 and other tracers. Please clarify.

The additional source of $CH_4$ is due to agricultural activities from the paddy fields in the monsoon season (esp. August-September) which is absent from October. Therefore we see the drop in $CH_4$ starting October. It is likely that the absence of rainfall after October is conducive to $CO_2$ accumulation and thus we see an increase in the mixing ratio of $CO_2$ thereafter. We have tried to clarify it in lines 432-433 as:

However, the reduction in ambient $CH_4$ after October could be due to reduced $CH_4$ emissions from paddy fields, which were high in August-September.

Page 15, lines 412-413 "which had > 2.5 ppm CH4 and > 450 CO2" It's likely that the increase in CO2 & CH4 is associated with advected signals from the North East and the East; however it is not clear how it can be interpreted as the advected airmass had the said values for CO2 and CH4 unless the study used any tracer transport and emission models. Wind direction and tracer concentration from the given site alone are not sufficient to conclude this. A clarification is needed here. Otherwise I recommend authors to remove this. Page 15, lines 414 "(not shown in Figure 4)" I encourage authors to show CO as well for the completeness of the interpretation.

We agree that in the absence of tracer transport and emission modeling, it is not sufficient to conclude the directionality of the advected signals. The sentence formulation also gave an impression that we are definitive about our conclusion. We have revised the lines 448-452. And CO has been included in Figure 4 as suggested by reviewer.

In the absence of tracer model simulations, the directionality of the advected air masses is unclear. Figure 4 shows that during these two months, $CO_2$ mixing ratios were particularly high ($> 450$ $CO_2$ and $> 2.5$ ppm $CH_4$) with the air masses coming from the Northeast-East (NE-E).

Page 15, lines 417 "high CH4 emissions" What about CO2?

The $CO_2$ level is lower during monsoon period in Asia due to high uptake of $CO_2$ by plants (high photosynthetic activities). This has been mentioned in lines 95-100 and 422-423 in the manuscript with reference.

ă˘A´lPage 17, line 455 "The westerly circulation (originated at longitude about 60E in 5 days back trajectories)" What does it mean? What's originated? Talking about model? Based on modeled trajectories?

The air mass back trajectory analysis is based on the study by Putero et al., 2015 for a site in the Kathmandu Valley for the period February 2013-January 2014. Using the Hysplit trajectory model, Putero et al (2015) clustered/grouped the 5 day back trajectories into a total of 9 clusters. The "westerly circulation" referred to in the manuscript is one of the dominant clusters (21.4%) observed by Putero. If spatially viewed on a lambert conformal projection, the majority of individual trajectories in the westerly circulation "or westerly cluster" originates in the region of $20-40^0$ N, $\sim 60^0$ E.

We acknowledge that the sentence was confusing. Please see the following correction which, we hope, provides a fairly detailed and clear explanation (lines 495-504).

To relate the influence of synoptic circulation with the observed variability in BC and $O_3$ in the Kathmandu Valley, 5-day back trajectories (of air masses arriving in the Kathmandu Valley) were computed by Putero et al., (2015) using the HYSPLIT model. These individual trajectories which were initialized at 600 hPa, for the study period of one year, and were clustered into nine clusters. Of the identified clusters, the most frequently observed clusters during the study period were the Regional and Westerly cluster or circulation (22% and 21%). The trajectories in the regional cluster originate within $10^o$ x $10^o$ around the Kathmandu Valley, whereas the majority of trajectories in this westerly cluster originated broadly around $20$-$40^0$ N, ~$60^0$ E. Putero et al (2015) found that the regional and westerly synoptic circulation were favorable for high values of BC and $O_3$ in the Kathmandu Valley.

Page 17, lines 475-476 "CO2 mixing ratios whereas CO shows an evening peak" It's surprising. Why is it so? Please clarify.

We meant to say that the $CO_2$ and $CH_4$ keep increasing over evening time until early morning while CO shows a peak coinciding with evening peak traffic hour and then drops. The decay in CO is more pronounced in monsoon and post-monsoon seasons. We have tried to clarify further in lines 526-534.

The gradual increase of $CO_2$ and $CH_4$ mixing ratios in the evening in contrast to the increase until evening peak traffic hours and later decay of CO may be indicative of a few factors. As pointed out earlier, after the peak traffic hours, there are no particularly strong sources of CO, especially in the monsoon and post-monsoon season. It is also likely that some of the CO decay is due to nighttime katabatic winds which replace polluted air masses with cold and fresh air from the nearby mountain (Panday and Prinn, 2009). As for the $CO_2$, the biosphere respiration at night in the absence of photosynthesis can add additional $CO_2$ to the atmosphere, which especially in the very shallow nocturnal boundary layer may explains part of the increase of the $CO_2$ mixing ratio.

Page 19, lines 519-521 "While the : : :.. other seasons" I couldn't follow how it's related. A clarification is highly needed. What are these other most CO sources mentioned here?

We have tried to make it clear and also included other CO sources in lines 578-581. The new sentences read as:

While the biosphere respires at night, which may cause a notable increase in $CO_2$ in the shallow boundary layer, most CO sources (transport sector, residential cooking) except brick kilns remain shut down or less active at night.

Page 20, lines 551-559 "Highest day : : :..Mauna Loa andWaliguan" It's lost. I see many assumptions here rather than convincing statements. What about biospheric activity and its seasonality? mesoscale transport mechanisms?

We acknowledge that the whole paragraph was not clear. We have tried to make it concise and clear and also included some of your suggestions. The paragraph has been revised as follows (lines 612-628):

The highest daytime minimum of $CO_2$ was observed in the pre-monsoon, followed by winter (Figure 6b). The higher daytime minimum of $CO_2$ mixing ratios in the pre-monsoon season than in other seasons, especially winter, is interesting. The local emission sources are similar in pre-monsoon and winter and the boundary layer is higher (in the afternoon) during the pre-monsoon (~1200 meters) than in winter (~900 meters) (Mues et al., 2017). Also, the biospheric activity in the region is reported to be higher in the pre-monsoon (due to high temperature and solar radiation) than winter (Rodda et al., 2016). Among various possible causes, transport of $CO_2$ rich air from outside the Kathmandu Valley has been hypothesized as a main contributing factor, due to regional vegetation fire  combined with westerly mesoscale to synoptic transport (Putero et al. 2015). In monsoon and post-monsoon seasons, the minimum $CO_2$ mixing ratio in the afternoon drops down to 390 ppm, this was close to the values observed at the regional background sites Mauna Loa and Waliguan.

Rodda, S.R., Thumaty, K. C., Jha, C. S. and Dadhwal, V. K.: Seasonal Variations of Carbon Dioxide, Water Vapor and Energy Fluxes in Tropical Indian Mangroves. Forests, 7, 35; doi:10.3390/f7020035, 2016.

Page 21, lines 577-579 "Overall, the : : :.. fire etc." Importantly it shares the transport mechanisms.

We would like to thank reviewer for raising the role of transport mechanism in this section and it has been added in lines 647-650. The revised sentence reads as follows:

Overall, the positive and high correlations between $CH_4$ and CO mixing ratios and between $CH_4$ and $CO_2$ in the pre-monsoon and winter indicate common sources, most likely combustion related sources such as vehicular emission, brick kilns, agriculture fire etc., or the same source regions (i.e. their transport from outside the Kathmandu Valley due to regional atmospheric transport mechanisms).

Pages 21-22, Sec. 3.5 I strongly recommend authors to remove the whole section. It is not at all straight forward, as assumed here, to determine the impact of emission sources and transport, based on concentration measurements from single site. It does not make any sense unless a dedicated further study is involved to justify the stated assumptions here. The section, as in the present shape, does not meet scientific reasoning; hence need to be removed.

We are in agreement with the reviewer that without any good supporting evidence, it can be difficult to accept the assumptions made. Thus, as suggested, the whole section has removed from the manuscript and the remaining sections are renumbered.

Page 23, lines 633-641 "Based on the : : :.. post-monsoon seasons" This could be a likely scenario. Have these interpretations been supported by any emission inventories available? It is also important to point out the associated uncertainties involved in separating different emission sectors based on this approach. Note that this approach cannot separate near and far field sources, different lifetimes of tracers etc.

Yes, we agree that the source identification based on $CO/CO_2$ ratio is indicative, not a definitive evidence. We apologize if we sounded definitive in our sentences and conclusion derived from the ratio analysis. We have also stated clearly in the paragraph that this method or the values associated with the source (Table 5) or our estimated values may have large uncertainty. A few additional lines have been added 680-682 to convey our cautious approach. Although we were unable to estimate the standard deviation of the ratio in Table 5, we have included the standard deviation of our calculated ratio in Table 6, and added useful statistics such as geometeric mean and geometric standard deviation and their upper and lower bounds. The whole paragraph is significantly modified, and as supporting evidence, we also have added emission source sectors in the Kathmandu Valley based on a high resolution emission inventory (Sadavarte et al., 2017, in preparation).. Here is the modified paragraph (lines 715-730):

Although ratio of $CO/CO_2$ is a weak indicator of sources and the mean ratio has large variance (See Table 6), the conclusions drawn from using Figure 8 and the above mentioned classification are not conclusive. The estimated $CO/CO_2$ ratio tentatively indicates that the local plume impacting the measurement site (Bode) from the north and east could be residential and/or diesel combustion. The estimated $CO/CO_2$ ratio of the local plume from the south and west generally falls in the 15-45 range, which could indicate emissions from brick kilns and inefficient gasoline vehicles. Very high ratios were also estimated from the south west during the post-monsoon season. Among other possible sources, this may indicate agro-residue open burning.

The emission inventory for CO identifies (aggregate for a year) residential, and gasoline related emission from transport sector (Sadavarte et al., 2017, in preparation). The inventory is not yet temporally resolved, so no conclusion can be drawn about the sources with respect to different seasons. From the 1km x1km emission inventory of the Kathmandu Valley for 2011, the estimated sectoral source apportionment of CO is residential (37%), transport sector (40%) and industrial (20%). The largest fraction from the residential sector is cooking (24%) whereas the majority of transport sector related CO in the Kathmandu Valley is from gasoline vehicles.

These observations can be useful as ground-truthing for evaluation of satellite measurements, as well as climate and regional air quality models.

Page 27, lines 747-749 "The analysis ... Kathmandu Valley", Please, remove this sentence. Note that this is not met here and the study only demonstrates the observational variations of GHGs in the study region.

We agree with the reviewer that this study focuses on observational variations of mainly $CH_4$ and $CO_2$. However, this study along with recent studies will help in addressing mitigation of the pollution in the study region. Thus, we have rephrased the sentence in lines 837-839as:

The overall analysis presented in the paper will contribute along with other recent measurements and analysis to providing a sound scientific basis for reducing emissions of greenhouse gases and air pollutants in the Kathmandu Valley.

Reference Please check. Formatting issues and sometimes journal details (or other important parts) are missing

We have re-checked the references and tried to resolve formatting and other issues in the section.

Table 4 Columns 4 & 5: What are these values? Looks like monthly values for Bode during Aug., and Sep. 2013; but then why are they different from corresponding $2^{nd}$ column of the table 4?

These are monthly values for Bode, which were simultaneously measured with another site at Chanban during August and September 2015. The information has been included in Table 4.

Table 6 Column 2: This "*" meant for?

We would like to thank reviewer for noticing this symbol. The meaning of the symbol has been included with a note at the bottom of the Table 6 as:

*The morning peak was one hour delayed in winter, thus the 8:00-10:00 period data was used in the analysis.

Figure 4 I didn't follow the fig. well. What I understood is that the plot shows the frequency of hourly mixing ratios w.r.t the frequency of prevailing wind direction. Did it also take into account the wind speed? Then what about different percentages shown? For example, does it mean that 5% of sample time in August, the wind was from NE and "CH4_corrected" is above 2.5 ppm in which less than 1% time (in my eyes), CH4 is in 3-3.3 ppm range? In that case, it's statistically difficult to say that the monthly enhancement is due to the polluted air masses from the NE and E. By the way, what are these "_corrected" values for CH4 and CO2? What about March-April scenario for CO2_corrected? Did two plots (Figs.2 and 3) use same set of master data, or any quality filtering had been done other than monthly averaging?

We are sorry for creating confusion by keeping "CH4_corrected" in the legend of Figure 4. $CO_2$ and $CH_4$ corrected means water corrected values of them, which we explained in section 2.2….. The $CH_4$ and $CO_2$ data used in the whole analysis (including Figure 2 and 3) are the same and they are water corrected values. We did not use any additional filtering while making these plots in the analysis. To avoid confusion, we have replaced $CH_4$_corrected and $CO_2$_corrected by $CH_4$ and $CO_2$ respectively in the legend of Figure 4.

[revised manuscript text omitted]

---

## Author Response (AR2)

**Response to Co-Editor's comments**

Dear Editor, we would like to thank you once again for your comments and suggestions which are useful for maintaining the quality of the manuscript and improve its write up. The authors' response and changes in the manuscript are presented in "blue" color.

Comments to the Author:

Additional technical corrections are needed prior to publication and are listed below by line number:

Thank you very much for pointing out these technical corrections which are important to improve the quality of the manuscript.

Lines 306, 311, and 313: Clarify what is meant by "changes" and "variabilities." As stated these are too vague and it is unclear what specific measures are being discussed. Perhaps "relative standard deviation" is the correct term?

We would like to apologize for creating confusion on changes and variabilities. We also agree with your suggestion using "relative standard deviation" instead of changes or variabilities. We have changed it throughout the manuscript.

Line 756: The directions "East-South/North East" is confusing. Please break this up into parts to omit the slash.

45: This comparison suggested that

213: iron, fabrics, etc.

215: 4 km to the west of Bode.

220: located approximately 4.5

234: that results in high sensitivity. (Omit precision here, as this is not affected by the path length.)

236: interest to reduce drift.

238: precisions for CO2…

244: data every 5 seconds and… corrected data for CO2

247: field measurements.

254: simultaneous measurements…

258: The Horiba…

265: A statistically…

271: deemed unnecessary.

272: Besides being highly…

292: from entering into…

293: an AWS….

299: 3.5 discuss the interrelation…

306: mixing ratio were higher…

327: spell out "ca."

344: Valley, the measurement…

348: (small-medium) and biomass…

356: The small differences…

357: the smaller number…

373: than in the pre-monsoon….

383: all seasons…

387: CH4 emissions…

390: brick kilns

396: suppresses emissions.

398: and, where relevant, reduction…

400: during high intensity

409: related to little or no rainfall

412: relatively low during…

426: increased by…

430, 432: East-Northeast (E-NE)

447: local sources…

462: emissions form the main…

464: higher values of all…

487: farm fires…

492: were coincident

498: diurnal patterns

506: decayed

509: may explain…

533: verb is needed to complete the clause following "and"

546: from the plains to the mountains

548: Although the CO…

565: Omit "Measurement in"

583: change in the mixing later

596: vegetation fires

771: seasons and wind directions suggested

Table 6 caption: and CO2 and their lower and upper bounds (LB and UB).

Table 5 caption: change "gram of gas" to "mass of gas"

Table 5 caption: ", except for the transport sector"

Table 1: use capitalization consistently in the headings.

Figure 9 caption: in all seasons except…

Thank you for providing detail technical corrections. We have incorporated all the technical corrections mentioned above in the revised manuscript. Furthermore, we have checked the ACP's manuscript preparation guidelines and style once more carefully, and consistently followed the style, for example units and time format, throughout the manuscript (for example, 10 % NOT 10%, 09:00 instead of 9 am).

**Seasonal and diurnal variations of methane and carbon dioxide in the Kathmandu Valley**

**in the foothills of the central Himalaya**

Khadak Singh Mahata[1,2], Arnico Kumar Panday [3,4], Maheswar Rupakheti[1,5*], Ashish Singh[1],

Manish Naja[6], Mark G. Lawrence[1,2]

[1] Institute for Advanced Sustainability Studies (IASS), Potsdam, Germany

[2] University of Potsdam, Potsdam, Germany

[3] International Centre for Integrated Mountain Development (ICIMOD), Lalitpur, Nepal

[4] University of Virginia, Virginia, USA

[5] Himalayan Sustainability Institute (HIMSI), Kathmandu, Nepal

[6] Aryabhatta Research Institute of Observational Sciences (ARIES), Nainital, India

*Correspondence to: M. Rupakheti (maheswar.rupakheti@iass-potsdam.de)

**Abstract**

The SusKat-ABC (Sustainable Atmosphere for the Kathmandu Valley- Atmospheric Brown

Clouds) international air pollution measurement campaign was carried out during December

2012-June 2013 in the Kathmandu Valley and surrounding regions in Nepal. The Kathmandu

Valley is a bowl-shaped basin with a severe air pollution problem. This paper reports measurements of two major greenhouse gases (GHGs), methane ($CH_4$) and carbon dioxide ($CO_2$), along with the pollutant CO, that began during the campaign and were extended for a year at the SusKat-ABC supersite in Bode, a semi-urban location in the Kathmandu Valley.

Simultaneous measurements were also made during 2015 in Bode and a nearby rural site (Chanban), ~25 km (aerial distance) to the southwest of Bode, on the other side of a tall ridge.

The ambient mixing ratios of methane ($CH_4$), carbon dioxide ($CO_2$), water vapor, and carbon monoxide (CO) were measured with a cavity ring down spectrometer (Picarro G2401, USA), along with meteorological parameters for a year (March 2013 - March 2014). These measurements are the first of their kind in the central Himalayan foothills. At Bode, the annual
average mixing ratios of $CO_2$ and $CH_4$ were 419.3 ($\pm6.0$) ppm and 2.192 ($\pm0.066$) ppm,
respectively. These values are higher than the levels observed at background sites such as Mauna
Loa, USA ($CO_2$: 396.8 $\pm$ 2.0 ppm, $CH_4$: 1.831 $\pm$ 0.110 ppm) and Waliguan, China ($CO_2$: 397.7 $\pm$
3.6 ppm, $CH_4$: 1.879 $\pm$ 0.009 ppm) during the same period, and at other urban and semi-urban
sites in the region such as Ahmedabad and Shadnagar (India). They varied slightly across the
seasons at Bode, with seasonal average $CH_4$ mixing ratios being 2.157 ($\pm0.230$) ppm in the pre-
monsoon season, 2.199 ($\pm0.241$) ppm in the monsoon, 2.210 ($\pm0.200$) ppm in the post-monsoon,
and 2.214 ($\pm$ 0.209) ppm in the winter season. The average $CO_2$ mixing ratios were 426.2
($\pm25.5$) ppm in pre-monsoon, 413.5 ($\pm24.2$) ppm in monsoon, 417.3 ($\pm23.1$) ppm in post-

[revised manuscript text omitted]